



# Projected increases in magnitude and socioeconomic exposure of global droughts in 1.5 °C and 2 °C warmer climates

Lei Gu[1], Jie Chen[1,2*], Jiabo Yin[1*], Sylvia C. Sullivan[3], Hui-Min Wang[1], Shenglian Guo[1], Liping Zhang[1,2], Jong-Suk Kim[1,2]

[1]State Key Laboratory of Water Resources and Hydropower Engineering Science, Wuhan University, Wuhan 430072, P. R. China

[2]Hubei Provincial Key Lab of Water System Science for Sponge City Construction, Wuhan University, Wuhan, China

[3]Department of Earth and Environmental Engineering, Columbia University, New York, NY 10027, USA

*Correspondence to*: Jie Chen (jiechen@whu.edu.cn); Jiabo Yin (jboyn@whu.edu.cn)

**Abstract:** The Paris Agreement sets a long-term temperature goal to hold global warming to well below 2.0°C and strives to limit to 1.5°C above preindustrial levels. Droughts with either intense severity or a long persistence could both lead to substantial impacts such as infrastructure failure and ecosystem vulnerability, and they are projected to occur more frequently and trigger intensified socioeconomic consequences with global warming. However, existing assessments targeting global droughts under 1.5°C and 2.0°C warming levels usually neglect the multifaceted nature of droughts and might underestimate potential risks. This study, within a bivariate framework, quantifies the change of global drought conditions and corresponding socioeconomic exposures for additional 1.5°C and 2.0°C warming trajectories. The drought characteristics are identified using the Standardized Precipitation Evapotranspiration Index (SPEI) combined with the run theory, with the climate scenarios projected by 13 Coupled Model Inter-comparison Project Phase 5 (CMIP5) global climate models (GCMs) under three representative concentration pathways (RCP2.6, 4.5 and 8.5). The copula functions and the most likely realization are incorporated to model the joint distribution of drought severity and duration, and changes in the bivariate return period with global warming are evaluated. Finally, the drought exposures of populations and





regional gross domestic product (GDP) under different shared socioeconomic pathways
(SSPs) are investigated globally. The results show that within the bivariate framework,
the historical 50-year droughts may double across 58% of global landmasses in a 1.5°C
warmer world, while when the warming climbs up to 2.0°C, an additionally 9% of world
landmasses would be exposed to such catastrophic drought deteriorations. More than
75 (73) countries' population (GDP) will be completely affected by increasing drought
risks under the 1.5°C warming, while an extra 0.5°C warming will further lead to an
additional 17 countries suffering from a nearly unbearable situation. Our results
demonstrate that limiting global warming to 1.5°C, compared with 2°C warming, can
perceptibly mitigate the drought impacts over major regions of the world.
**Keywords:** Global warming; Drought; Copula function; Most likely scenario;
Socioeconomic exposures

## 14   1. Introduction

Climate warming mainly due to greenhouse gas emissions has altered the global
hydrological cycle and resulted in more frequent and persistent natural hazards such as
droughts, which have imposed considerable economic, societal, and environmental
challenges across the globe (Handmer et al., 2012; Chang et al., 2016; EM-DAT 2017).
With the aspiration to mitigate these adverse consequences, the Paris Agreement
proposed to cut greenhouse gas emissions for holding the increase in global temperature
to well below 2.0°C and pursuing efforts, limiting the warming to 1.5°C above pre-
industrial levels (UNFCCC, 2015). Regardless of the socioeconomic and technological
achievability of the Paris Agreement goals, portraying the drought evolution with
different warming trajectories would provide valuable information and references for
mankind to enable appropriate adaptation strategies in a warmer future.

26       To examine the sensitivity of drought risks with different warming targets, numerous

approaches have emerged. One way is to employ a set of ensemble simulations
produced by a single coupled climate model (e.g., Community Earth System Model,





CESM), which is designed specifically to perform the impact assessments at a near-
equilibrium scenarios of 1.5°C or 2°C additional warming (Sanderson et al., 2017;
Lehner et al., 2017). This single model type cannot reflect the structural uncertainty of
climate models, which is important in impact assessments, and thus raises doubts about
the robustness of such drought condition assessments (Liu et al., 2018). Emerging
modeling efforts such as the "Half a degree Additional warming, Projections, Prognosis
and Impacts" (HAPPI) model inter-comparison project provided a new dataset with
experiments designed to explicitly target impacts of 1.5°C and 2°C above preindustrial
warming (Mitchell et al., 2016). However, the HAPPI employed prescribed
climatological sea surface temperatures and could not consider the internal variability
of ocean-atmosphere circulation, which is crucial in physically simulating climatic
variability and persistence (Seager et al., 2005; Routson et al., 2016). Current studies
usually utilize CMIP5 climate models to project climate scenarios under different RCPs,
identify the time period for a warming target and then examine the drought conditions
associated with different levels of global warming. For instance, Su et al. (2018) used
13 CMIP5 models based on RCP 2.6 and RCP 4.5 to compare the drought conditions
for two warming targets over China and reported tremendous losses will emerge even
under the ambitious 1.5°C warming target.
These prevailing tides of literature almost reach a consensus that, with higher
saturation threshold and more intense and frequent dry spells driven by rising
temperatures, drought conditions would considerably worsen in many regions of the
world (Mitchell et al., 2016; Liu et al., 2018). The potentially devastating impacts of
more severe drought conditions on society raise considerable concerns, motivating a
number of global socioeconomic assessments of future drought change impact (e.g.,
Below et al., 2007; Schilling et al., 2012). For instance, Liu et al. (2018) investigated
global drought evolution and corresponding population exposures in additional 1.5°C
and 2°C warming conditions using a set of CMIP5 models under RCP 4.5 and RCP 8.5.
Naumann et al. (2018) assessed the development of drought conditions across the world
for different warming targets in the Paris Agreement. These studies concluded that there



are considerable benefits for the environment and society of limiting warming to 1.5°C
relative to 2.0°C, although 1.5°C warming still implies a substantial challenge for global
sustainable development. However, most previous socioeconomic assessments (e.g.,
Peters, 2016; Park et al., 2018; Liu et al. 2018) have focused on a static socioeconomic
scenario, probably due to data constraint. These studies cannot capture the dynamic
nature of population and assets over time, that has been identified as crucial for
simulating realistic societal development path (Smirnov et al., 2016). Recently, five
Shared Socioeconomic Pathways (SSPs) have been proposed, providing a more
reasonable dataset to characterize a set of plausible alternative futures of societal
development with consideration of climate change and policy impacts over the 21st
century (Leimbach et al., 2017). To date, the SSPs have not yet been incorporated into
the drought impact assessments with warming at the global scale.

13         More importantly, among existing global drought impact assessments, especially

those targeting different warming levels proposed by the Paris Agreement, drought
variables such as severity and duration are usually separately investigated through
probability modelling and stochastic theories (e.g., Sanderson et al., 2017; Lehner et al.,
2017; Su et al., 2018). Knowing that droughts are multifaceted phenomena (Xu et al.,
2015; Tsakiris et al., 2016) usually characterized by duration and severity, univariate
frequency analysis is unable to describe the probability of occurrence for the drought
events physically and may lead to underestimation of drought risks and societal hazards.
For instance, droughts with a moderate severity but a long persistence are seldom
identified as severe events in univariate analysis; nevertheless, they may pose
substantial socioeconomic losses because of rapid stored water depletion and low
resilience to subsequent droughts (Lehner et al., 2017). Therefore, there is an urgent
necessity to incorporate the joint modeling of multiple drought features into impact
assessments (Genest et al., 2007; Liu et al., 2015). The copula function that shows good
feasibility of marginal distributions in modeling inter-correlated variables has been
introduced in multivariate analysis for droughts (e.g., Wong et al. 2013; Zhang et al.
2015; Ayantobo et al., 2017). However, to the authors' knowledge, no previous work



links the high interdependence of drought characteristics to a global impact assessment
under different warming levels.

3        In the multivariate framework, selection of variable combinations along the

quantile curve poses a new challenge, as the choice of the joint return period (JRP) leads
to infinitely many such combinations. To meet the needs of infrastructure design and
adaptivity, many researchers (e.g., Chen et al. 2010; Li et al. 2016; Zscheischler et al.,
2017) have assumed that the correlated variables have the same probability of
occurrence under a given JRP, which is called the equivalent frequency combination
(EFC) method. Despite the fact that the EFC method has low calculation complexity,
the statistical and theoretical basis of the equal frequency assumption is questionable
(Yin et al. 2018a). To develop a more rational design for a multivariate approach, a
novel concept of "most likely design realization" to choose the point with the highest
likelihood along the quantile curve has been proposed in frequency analysis (Salvadori
et al. 2011; Yin et al. 2019). It would be very important to evaluate and characterize
these different likelihoods of drought events in bivariate drought impact assessment
under a warming climate.

17        In this study, under a bivariate framework, we quantify changes in global drought

conditions and socioeconomic exposure with additional levels of 1.5°C and 2.0°C
warming. The drought characteristics are identified using the Standardized
Precipitation Evapotranspiration Index (SPEI) combined with the run theory and with
climate scenarios simulated by 13 CMIP5 GCMs under three RCPs (RCP2.6, 4.5, and
8.5). The copula functions and most likely realization are incorporated to model the
drought severity and duration concurrently, and changes in the bivariate return period
with global warming are systematically investigated. Finally, the drought exposures of
populations and regional GDP under different shared socioeconomic pathways (SSPs)
are assessed globally.




## 2. Materials and Method

### 2.1 Climatic and socioeconomic scenarios

Climate projections are based on ensemble runs (r1i1p1) by 13 models from CMIP5 (Table 1), covering the period 1976-2100 under three RCPs (i.e., RCP 2.6, 4.5, and 8.5). Ten climate variables were used in this study. Specifically, 9 out of the 10 variables were applied for the calculation of potential evapotranspiration (PET). These 9 variables include: surface mean air temperature, surface minimum air temperature, surface maximum air temperature, surface wind speed, relative humidity, surface downwelling longwave flux, surface upwelling longwave flux, surface downwelling shortwave flux, and surface upwelling shortwave flux. The 10th variable is the precipitation. Then the calculated PET and GCM-simulated precipitation were employed to calculate drought indices. The PET was initially calculated at the daily scale. Then both the daily scale PET and precipitation were aggregated to the monthly scales, and bilinearly interpolated to a spatial resolution of $1.0° \times 1.0°$ on latitude and longitude for each model simulation.

To assess the exposures of populations and assets to droughts, which will eventually lead to higher drought losses in the future, instead of using a static socioeconomic scenario as many studies have (e.g., Hirabayashi et al., 2013; Smirnov et al., 2016), we employ the spatially explicit global shared socioeconomic pathways (SSPs). This dataset includes gridded population and GDP data under five SSPs, covering the period 2010-2100 at a spatial resolution of $0.5°\times0.5°$ (Jiang et al., 2017; 2018; Su et al., 2018; Huang et al., 2019). It involves a sustainable scenario (SSP1), a pathway of continuing historical trend (SSP2), a strongly fragmented world (SSP3), a highly unequal world (SSP4), and a growth-oriented world (SSP5). Among combinations of different RCP trajectories and socioeconomic pathways, some SSP-RCP combinations are unlikely to occur, e.g., SSP3-RCP2.6 and SSP1-RCP8.5 (Jones et al., 2016). Considering the socioeconomic challenges for mitigation along different



development paths, the RCP2.6 scenario is associated with SSP1, which will face a
lower challenge of mitigation in the future. The RCP4.5 scenario is associated with the
SSP2, while the highest emission scenario RCP 8.5 is associated with the SSP5, by
which a relatively higher challenge is expected under foreseeable warming conditions
(Samir et al., 2017).
**2.2. Definition of a baseline, 1.5°C and 2°C global warming**
The sensitivity of annual global temperature to climate variability significantly varies
in models and RCPs. Therefore, the time period with additional global warming of 1.5°C
and 2°C with respect to pre-industrial conditions also varies between different climate
scenarios. Here, the time periods for different global warming levels are determined
using the 30-year running-mean of multi-model ensemble mean of global-mean surface
air temperature, following previous studies (Vautard et al., 2014; Su et al., 2018). We
first select a baseline period of 1976-2005, during which the observed global average
temperature was approximately 0.46-0.66°C warmer than pre-industrial condition
(IPCC, 2018). This reference period is widely adopted for climate impact assessment
(e.g., Vautard et al., 2014), and we set the warming degree during baseline period as
0.51°C; hence the 1.5°C and 2.0°C warming targets are determined by additional
warming of 0.99°C and 1.49°C, respectively. For each RCP, we define the 1.5°C and 2°C
warmer worlds during which the moving 30-year period with global warming closely
approximates to the corresponding warming levels (see Fig. 1).
**2.3 Drought indices and event identification**
**2.3.1 Standardized Precipitation Evapotranspiration Index**
The drought condition is quantified with the SPEI developed by Vicente et al. (2010),
which has been widely adopted in characterizing drought conditions (e.g., Ayantobo et
al., 2018; Wen et al., 2018). The SPEI quantifies the extent of atmospheric water surplus
and deficit relative to the long-term average condition by standardizing the difference
between precipitation and potential evapotranspiration (PET). The SPEI with 3-month





time scale (SPEI-3) is used in this study because it captures well the shallow soil
moisture available to crops and reflects seasonal water loss processes (Yu et al., 2014).

3        The PET is first calculated using the Penman-Monteith approach suggested by the

Food and Agriculture Organization of the United Nations (FAO) (Allen et al., 1998):

$$PET = \frac{0.408\Delta\left(R_n - G\right) + \gamma\frac{900}{tmean + 273}u_2\left(e_s - e_a\right)}{\Delta + \gamma\left(1 + 0.34u_2\right)} \qquad (1)$$

where $\Delta$ is the slope of saturation vapor pressure vs. air temperature curve (kPa /°C)
and is calculated by:

$$\Delta = 4098 \times \frac{0.6108 \times e^{\frac{17.27 \times tmean}{tmean + 237.3}}}{tmean + 237.3} \qquad (2)$$

where $tmean$ is the surface mean air temperature (°C). $R_n$ is the net radiation (MJ/m$^2$/day)
and is calculated by:

$$R_n = [rsds - rsus - (rlus - rlds)] * 10^6 * 3600 * 24 \qquad (3)$$

where $rsds$ and $rsus$ ($rlds$ and $rlus$) are surface downwelling and upwelling shortwave
flux (surface downwelling and upwelling longwave flux), respectively (w/m$^2$). $G$ is the
soil heat flux (MJ/m$^2$/day) and is close to zero at the daily scale. $\gamma$ is psychometric
constant (kPa/°C) and is calculated by:

$$\gamma = 0.665 \times 10^{-3} \times P \qquad (4)$$

where $P$ is the atmospheric pressure (kPa). $u_2$ is the wind speed at 2m height (m/s),
transferred from:

$$u_2 = 4.87 \times u_{10} / \ln(67.8 \times 10 - 5.42) \qquad (5)$$

where $u_{10}$ is the surface wind speed at the 10m height simulated by GCMs. $e_s$ and $e_a$ are
saturation and actual vapor pressure (kPa), respectively:

$$e_s = 0.6108 \times e^{\frac{17.27 \times tmp}{tmp + 237.3}} \qquad (6)$$

$$e_a = \frac{rhs}{100} \times e_s \qquad (7)$$

where $rhs$ is the relative humidity (%), and $tmp$ is temperature (i.e., daily maximum and





minimum air temperature). Due to the non-linearity of eq. (6), it would be more
appropriate to apply the average saturated vapor pressure derived from the daily
maximum and minimum air temperature.
The widely used Log-logistic distribution is employed for fitting the 3-month
deficit of precipitation and PET (P-PET) (Touma et al., 2015):
$$F(\text{x}) = [1 + (\frac{\alpha}{x - \lambda})^{\beta}]^{-1} \qquad (8)$$
where, $F(x)$ denotes the cumulative distribution function; $\alpha$, $\beta$ and $\lambda$ represent shape,
scale and location parameters, which are estimated by the maximum likelihood method
(Ahmad et al., 1988).
The SPEI-3 can then be derived by standardizing the $F(x)$ into a standard normal
function with a transforming function $\Phi^{-1}$ as follows:
$$SPEI_{-3}(x) = \Phi^{-1}(F(x)) \qquad (9)$$
**2.3.2 Drought event identification**
After calculating the SPEI-3 for global terrestrial grid cells, we derive the drought
duration, intensity, and severity using the run theory for the reference and the 1.5°C and
2°C warmer worlds. The run theory proposed by Yevjevich et al. (1967) is a useful and
objective method for drought event identification, where a run represents a subset of
time series, in which SPEI-3 is either beneath (i.e., negative run) or over (i.e., positive
run) a fixed threshold. A run with SPEI-3 that continuously stays below -0.5 is defined
as a drought event (Mishra et al., 2010; Zargar et al., 2011), which generally includes
drought characteristics of duration and severity. The persistent time period during a
drought event is further defined as the drought duration, while drought severity
(dimensionless) is defined as a cumulative deficit below -0.5.
**2.4 Bivariate return period and most likely realization method**
Previous studies usually independently examined the change either in drought duration
or severity under climate warming, neglecting the multiplex nature of droughts
(Naumann et al., 2018). This study jointly models drought duration ($D$) and severity ($S$)





via the copula function, which is versatile for describing dependent hydrological
variables due to its good flexibility of marginal distributions. The widely-used Gamma
distribution was adopted for fitting drought variables in each grid over the globe, and
we selected the Gumbel Copula to model the joint distribution of drought duration and
severity. Within the copula-based approaches, different definitions of joint return
periods (JRPs) have been proposed, such as OR, AND, Kendall, dynamic, structure-
based return periods (Yin et al., 2019). Among these, the OR case ($T_{or}$) is usually
adopted in drought occurrence assessment (Zhang et al., 2015):

$$T_{or} = \frac{E_l}{1 - F(d,s)} = \frac{E_l}{1 - C[F_D(d), F_S(s)]} \qquad (10)$$

where, $E_l$ represents the expected inter-arrival time of drought events, the joint
distribution $F(d, s)$ could be described by a copula function $C[F_D(d), F_S(s)]$; $F_D(d)$ and
$F_S(s)$ indicate the marginal distribution functions of $D$ and $S$, respectively.
Under the bivariate framework, the choice of an appropriate $T_{or}$ leads to infinite
combinations of drought duration and severity. The drought events along the $T_{or}$-level
curve are generally not equivalent in terms of environmental and societal consequences,
and hence the likelihood of each event must be taken into consideration when selecting
appropriate joint quantiles. In this paper, the most likely realization method (Salvadori
et al., 2011; Yin et al., 2019) is used to choose the drought scenario with the highest
likelihood along the $T_{or}$ -level isoline. For a given $T_{or}$, the most likely combination point
among all possible events can be derived by the following formula (Gräler et al., 2013):

$$\left\{ \begin{array}{l} (d^*, s^*) = \arg\max f(d,s) = c[F_D(d), F_S(s)] f_D(d) f_S(s) \\ C(F_D(d), F_S(s)) = 1 - E_l / T_{or} \end{array} \right\} \qquad (11)$$

where, $f(d,s)$ represents the joint probability density function of drought duration and
severity,  $c[F_D(d), F_S(s)] = dC(F_D(d), F_S(s)) / d(f_D(d)) \, d(f_S(s))$  indicates the density
function of copula;  $f_D(d)$ and  $f_S(s)$ are probability density functions of drought
duration and severity, respectively. Due to the complexity of deriving analytical
solutions in Eq. (5), the harmonic mean Newton's method (Yin et al., 2018a) is applied



to estimate the most likely realizations.
**2.5 Calculation of socioeconomic exposure under warmer condition**
To calculate the socioeconomic exposures by droughts in different warming
environments, we evaluate the change of drought occurrence frequency in a bivariate
context. Firstly, we estimate the bivariate quantiles of drought duration and severity
(i.e., most likely realization) under one given JRP during the historical period. As the
50-year drought events usually gained great attention by the scientific community and
socio-climatic policymakers (Zhang et al. 2015; Naumann et al., 2018), we adopt this
level as a reference for assessing possible drought implications. With the historical 50-
year bivariate quantiles, we can recalculate the joint occurrence frequency under future
additional 1.5°C and 2.0°C warming conditions, respectively. It can be inferred that
areas with a JRP lower than 50 years are projected to suffer from more severe drought
conditions. To explicitly assess the drought risk changes from 1.5°C to 2.0°C warming
climates, we estimate the ratio of the recalculated recurrence frequency between these
two warming periods, where those areas with a less than 1.0 ratio are projected to be
exposed to worrisome drought conditions.

17        To evaluate socioeconomic implications of drought with additional warming, we

record the population and GDP in those areas with more severe drought conditions and
define them as exposures by increasing drought risks. As previously stated, we consider
the dynamic nature of socioeconomic development pathways by employing different
SSPs, and used the multi-year average populations and GDPs during 30-year periods
determined by different warming levels. After estimating the socioeconomic exposures
for each GCM simulation, we use the multi-model ensemble mean as an indication for
each grid cell to reduce model bias. Note that we select three RCPs and corresponding
SSPs under two warming targets so that the analysis is performed on six scenarios.



# 1  3. Results

**3.1 Projected changes in dryness**
We first examine changes in the mean and standard deviation of SPEI-3 from the
historical reference period (1976-2005) to the 1.5℃ warmer worlds (Fig. 2), indicated
by the multi-model ensemble mean results. We find that mean SPEI-3 decreases at the
global scale (across 85% of the land areas, excluding Antarctica), except in very limited
regions at high-latitude areas (e.g., Siberia in Russia) where it exhibits a slight increase.
The descending changes in the mean SPEI-3 imply that, over the majority of the globe,
the probability distribution function of SPEI-3 would shift towards lower values and
hence more severe dryness. Particularly, dramatic decreases combined with strong
model agreement (in terms of sign of change) are presented in Southern America,
Australia, and Northern Africa. This may be attributed to higher evaporative demands
and more frequent and persistent dry spells associated with rising temperatures
(Naumann et al., 2018). On the other hand, we also observe an increase in the standard
deviation of SPEI-3 with additional 1.5℃ warming, particularly in Northern Africa and
Southwestern Asia. As the SPEI-3 follows the standard normal distribution, the
increasing standard deviation means more variability in dryness, which hinders
resilience efforts in a 1.5℃ warmer world. These changes are consistent under three
different RCPs, indicating the robustness of this globally drier future.
How would the dryness pattern change from 1.5℃ to 2.0℃ warming climates? A
progressive descending change in mean values of SPEI-3 is observed across 58% of the
land surface with the global mean temperature increasing between 1.5℃ and 2.0℃,
although several high-latitude regions (i.e., Russia, Canada) show an insignificant
opposite change. This may be mechanically explained by thick clouds in these regions
that strengthen the reflectance of shortwave radiation and limit the increase of latent
heat flux as well as evapotranspiration, thus contributing to the mitigation of
atmospheric aridity (Huang et al., 2017). For the change in the standard deviation of
SPEI-3, we find that increases occur over continental regions almost globally,



accompanied by minor spatial variability. Overall, the climatic metric SPEI-3 shows a
strong negative response to the warming climate, suggesting that dryness will intensify
in a future warming world.

**3.2 Projected changes in drought characteristics**

Fig. 4 shows the relative change of global drought duration and severity derived from
SPEI-3 in the 1.5°C warmer world relative to the historical period under three different
RCPs. The drought duration is projected to slowly prolong with warming across 78%
of the land surface, and 44% of land areas has an increase of higher than 10%, although
the change is not significant in Russia and Sahel areas. The drought severity shows a
much more pronounced rise globally, with significant increases (exceeding 50%) over
46% of global landmasses. Moreover, several regions experience compound increases
(with strong model agreement) in both drought severity and duration, such as Southeast
Asia, Mediterranean, Southern Africa, Southern North America, and South America,
suggesting an urgent need to increase societal and environmental resilience to a
warming climate there. In the tropics and high-latitudes areas, the drought severity is
projected to increase while the duration will decrease. In these regions, mitigation
strategies should target short, intense bursts of drought.
When the global temperature rises from additional 1.5°C to 2.0°C warming, the
world would experience more severe drought conditions, with a further increase in
drought severity accounting for 75% of the land surface (differences in effects between
the 1.5°C and 2.0°C warming levels) and a persistent lengthen in duration across 58%
of the land areas (Fig. 5). Similar to the changing pattern from baseline to a 1.5°C
warming climate, the drought severity shows a more rapidly increasing rate than
drought duration globally under the 2.0°C warming world. Comparing the 2.0°C to the
1.5°C warming condition, the increase in drought severity is greater than 10% over 35%
of the land areas, while only 8% of the land areas show such an increase (>10%) in
drought duration. This drought-prone condition is more severe in several regions such
as Mediterranean regions, South Africa and South America, posing large challenges for
existing socio-hydrological systems there.



To explicitly investigate the changes of drought characteristics under warming
conditions, we also show statistics of drought frequency, duration and severity in the
historical period and future additional warmer worlds in violin plots (Fig. 6), in which
the distributions comprise drought characteristics across all land pixels of the multi-
model ensemble mean results. The violin plots (Hintze et al., 1998) consist of a boxplot
inside and an outside violin shape which displays the probability distribution of drought
characteristics. Apparently, the drought frequency based on SPEI-3 is also projected to
pronouncedly lengthen under three RCPs, accompanied by large variability capturing
by the kernel density estimation in Fig 6. This rapid increasing tendency also holds true
for drought duration and severity, and extreme conditions are projected to occur more
frequently under warming climates. For example, the 90% uncertainty range of drought
duration (severity) increases from 2.2-6.5 months to 1.8-7.8 months (from 2.1-6.6 to
2.0-12) under 2.0°C warming climate relative to the historical period.
**3.3 Projected changes in drought risks**
As evidence is accumulating that high-impact events are typically multivariate in nature
(Zhang et al. 2015; Ayantobo et al., 2017), we now consider a deeper focus on changes
in drought severity and duration within a bivariate framework under different warming
levels. Using the copula-based approach in Section 2.4, we show the median projected
change of the historical 50-year drought conditions over multi-model ensembles under
1.5°C warming climate (Fig. 7). Generally, in regions with a substantial increase in
drought duration and severity (Fig. 5), the 50-year drought events exhibit a rapid
increase in occurrence with warming. More than 88% of global landmasses will be
subject to more frequent historical 50-year droughts, and the frequency of such severe
droughts would double over 58% of the global land surface. For most areas of South
America (except for the zone around the equator), Northeastern America, Central, and
West Asia, and northwest China, the historical 50-year droughts are projected to occur
2 to 10 times more frequently under the ambitious 1.5°C warming level. Regions with
a lower frequency of historical 50-year drought event indicate a reduction in drought
risks, which are only limited in Siberia, India Peninsula, and Alaska.




To closely assess the drought conditions with an extra 0.5°C warming, we derive
the ratio of adjusted 50-year return period between 2.0°C and 1.5°C warming worlds
(Fig. 8). In regions with a ratio of less than 1.0, the present drought events are projected
to occur more frequently under the half a degree additional warming, which accounts
for 71% of continental areas. In addition, the frequency of the historical 50-year
droughts would double across 67% of the global landmasses under the 2.0°C warming
level. That is, 9% increase of the world land areas compares to the 1.5°C warming level
(i.e., 58%). Although over some regions such as northern Canada and Eastern Asia, the
occurrence of the extreme droughts will be less frequent to some degree, strong rises in
recurrence frequency with warming are projected to dominate large parts of Europe, the
southern United States, Australia, South America, Northern Africa, and the
Mediterranean.
**3.4 Population and GDP exposure from increasing drought risks**
To understand the socio-economic influences induced by increasing drought risks (here
defined as more frequent historical 50-year events), we combine the drought projection
with population and GDP information based on SSPs, and estimate exposures by
droughts in the 1.5°C and 2.0°C warmer worlds. Globally, three RCPs suggest a
consistent projection that large percentages of population and GDP will be exposed to
increasing drought risks. In more than 67 (140) countries, 100% (50%) of both
populations and GDPs are exposed to more severe droughts under the 1.5°C warming
target (Fig. 9). The two socioeconomic factors of GDP and population are highly
correlated (O'Neill et al., 2014). Economically prosperous regions are associated with
higher population and immigration (Fig. S1); thus the drought-affected GDP exposures
usually exhibit similar changing pattern with the population.

25       In regions with low GDP and population density, even when total socioeconomic

exposures to droughts seem small, droughts can still cause fatal and destructive losses
for those countries if their drought resilience is poor. To give a fairer and more impartial
assessment of droughts' socioeconomic consequences, we define and assess the fraction
of drought-affected population (or GDP) divided by total population (or total GDP)

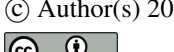



based on different countries in a 1.5°C warming world. With this national assessment
method, we see interesting results (Fig. 9). For example, the United States and China
are no longer the most drought-affected countries, while 100% of the population and
GPD in Mexico, Southern Europe, Middle, and Southern Africa, and Mediterranean
regions (i.e., Turkey, Ukraine) are projected to experience more severe drought,
suggesting large policy challenges there. To illustrate the consequences of limiting
warming to 2.0°C above the preindustrial levels, we also calculate the socioeconomic
exposures under three RCPs (Fig. 10) and the differences in percentage between the
1.5°C and 2.0°C warming levels (Fig. S2). Most regions of the globe are projected to
exhibit a generally increasing fraction (relative to 1.5°C warming) in populations and
GDPs (except for Central Africa and East Asia). To be specific, under the extra half-
degree warming, an additional 17 countries are projected to exhibit a 100% fraction in
socioeconomic exposure. More than 10 countries would experience a 30% increase in
population and GDP exposure if the global warming level increased from 1.5°C to 2.0°C.
These increases illustrate the benefit of holding global warming to 1.5°C instead of 2°C,
particularly for the mitigation of population and GDP exposure to drought.

**17    3.5 National assessment of socioeconomic exposure in typical countries**

The drought risks and socioeconomic exposures under warming climates exhibit large
spatial variability, which motivates a more systematic and in-depth assessment on
national scales, particularly for the countries vulnerable to droughts. Therefore, we
investigate more thoroughly the drought-affected land fractions (Figs. 11-12) and
corresponding socioeconomic exposure (Figs. S3-4) in eight hotspot countries spanning
different socio-climatic regions: Argentina, Australia, Canada, China, United States,
South Africa, Brazil, and Mexico.
For assessment at the national scale, spatially aggregating mean changes are more
helpful than per-grid cell changes to indicate the risk of a particular land fraction being
impacted by climate change (Fischer et al., 2013; Lehner et al., 2017). The land
fractions of each grid cell are binned and plotted against the change of drought return
period (relative to historical 50-year drought) (Figs. 11-12). The bin number is fixed to



20 groups for the eight example countries. In a 1.5°C warming world (Fig. 11), these
spatially aggregated changes explicitly show a significant increase in drought risks over
these hotspot countries, with more than 90% of grid cells projected to suffer from more
frequent droughts.
Nevertheless, we still observe a difference between the tropics and extratropical
regions. The increasing drought risks are more profound in tropical regions (e.g.,
Mexico and Brazil) than those over the high-latitude country (e.g., Canada). For
instance, in a 1.5°C warming world, more than 85% of the grid cells (associated with
around 65%-97% of the national populations and GDPs) over Mexico and Brazil could
be exposed to the historical 50-year drought every 20 years. This pronounced increase
in drought risks over tropical countries may be attributed to an oceanic forcing that
favors the formation of deep convection over the ocean and thus weakened the
continental convergence associated with the monsoon (Giannini et al., 2013). This
finding suggests that the tropics may confront more severe, frequent droughts and worse
socioeconomic influences (Figs. S3-4) under a warming climate. When the additional
warming target rises up to 2.0°C, drought conditions worsen over all these example
countries (Fig. 12). The increase in drought risks is still more pronounced in the tropical
countries. More than 90% of the grid cells (associated with around 90%-100% of the
national population and GDP) across Brazil and Mexico will experience drought
frequency double that of the historical 50-year drought.
Overall, increasing drought risks under warming climates can cause major
challenges for sustainable development and existing infrastructure systems, while
ambitiously limiting warming to 1.5°C would substantially mitigate future drought risks
and corresponding socioeconomic exposures.
**4. Discussion**
Among the warming-induced hydrological changes, one of the most definitive and
detectable changes is the simultaneous increase of precipitation and evaporative
demand, which are governed by the Clausius-Clapeyron relationship (Scheff et al.,





2014). Observations and model simulations have reported a variety of scaling rates
between precipitation and global temperature, where the daily and hourly precipitation
extremes (i.e., 99[th] / 95[th] percentile precipitation) usually exhibit a sub C-C scaling at
regional scales, accompanied by spatial and decadal variability (Yin et al. 2018b). For
global average precipitation, however, most climate models project an increase of 1-3%
per degree warming (Liu et al., 2013). This deviation from the C-C relation law is due
to a global radiative energy constraint (Held et al., 2006) and atmospheric moisture
limitation by decreasing relative humidity and increasing the potential for intense
tropical and subtropical thunderstorms under warming (Muller et al., 2011; Yin et al.
2018b). Potential evapotranspiration, on the other hand, is predicted to increase by 1.5-
4 % per degree warming (Scheff et al., 2014; Naumann et al., 2018). Therefore, we
expect climate warming to lead to a general intensification of drought conditions, as the
drying of the surface is enhanced with water scarcity. This is confirmed by the
decreasing SPEI-3 and significantly increasing drought severity and duration with
warming globally found here (Figs. 2-8).
Different threshold values in identifying a drought event may cause disparities
regarding drought risk changes and may challenge the robustness of our results.
Generally, the threshold value usually ranges between -1 and 0 (Xu et al., 2015;
Ayantobo et al., 2017, 2018; Yuan et al., 2017; Jiao et al., 2019). Herein, the threshold
of -0.5 is employed to identify droughts varying from mild to extremely dry levels
(Table 2, Chen et al., 2018), which has been widely adopted in drought-related studies
(Liu et al., 2015; Xiao et al., 2017; Chen et al., 2018). The inclusion of minor drought
events can enlarge the sample size in bivariate frequency analysis and thus circumvents
the problem of insufficient samples. Moreover, to verify the robustness of our results,
we also use the -0.8 threshold to serve as a comparison. Relevant results are shown in
Figs. 13-15. Fig.13 displays comparisons of distributions comprising drought
characteristics (i.e. drought frequency, drought duration and drought severity) across all
land pixels between using the -0.8 and -0.5 as the threshold. Figs. 14-15 show
comparisons of projected changes in joint 50-year return periods of droughts between



using the -0.8 and -0.5 as the threshold under different warming levels. As shown in the
figure (Fig.13), drought characteristics tend to slightly decrease across different periods.
However, future drought risk changes as indicated by the 50-year joint return period
deriving from the -0.8 threshold are similar to those from the -0.5 threshold (Figs. 14-
15). This confirms the conclusions of our study.
Although aggravated drought risks are projected globally, the changing patterns
exhibit large spatial variability, with more significant increases over mid-latitudes and
tropical regions than those over high-latitude landmasses. It should be noticed that
regions (e.g., the Mediterranean, Southern Africa, Southern North America) with large
projected changes generally display strong model agreement (in terms of sign of
change), which implies high confidence in these drought prone areas. Conversely,
substantial model uncertainty of drought projections is particularly clear for regions
with small changing amplitudes, as indicated by weak model agreement (e.g.,
Southeastern Asia and Russia).
Moreover, socioeconomic exposure (i.e., population and GDP) under different
warming levels is investigated in this work. Generally, drought conditions and
population (GDP) both contribute to the exposure change. In this study, we mainly
focus on the consequences derived from drought risk changes under different warming
levels. Accordingly, the exposure is defined as the number of people (GDP) being
exposed to areas where the bivariate drought risks increase under the warming climate.
The results indicate that drought risks represented by the joint return period will
significantly increase under the 1.5°C warming level and thus lead to severe impacts on
the population (GDP). Furthermore, an extra 0.5°C warming will result in increasing
drought risks, and at the same time, with ascending population (GDP), the exposure
risk will become more awful. Though not all the land areas (71% of global landmasses)
show increasing drought risks when the warming increases from 1.5°C to 2.0°C, a
further 9% increase in population (119% increase in GDP) will result in a greater
increase in the exposure and subsequently bring about more unbearable socio-economic
consequences. Extracting contributions from population (GDP) and drought risk



changes to the exposure variations is beyond the scope of this study. However, to better
serve for mitigation and adaptation strategies, there is a need to systematically partition
their relative contributions in future studies.
For example, 100% of the population in tropical regions like Brazil and Mexico
would be affected by increasing drought risks. Indeed, our finding that the tropical and
mid-latitude regions, where the vast majority of global population resides, would bear
the greatest drought risks should be precautious under the foreseeable warming future.
Previous studies have reported that the increases in El Niño frequency (Xie et al., 2010),
an extension of Hadley cell (Lu et al., 2007), and poleward moisture transport by
transient eddies (Chou et al., 2009) under warming all contribute to the drying tendency
in tropics; however, our work does not quantitatively examine these underlying physical
mechanisms behind the spatial variability due to paucity of data.
Besides the spatial variability of drought conditions and socioeconomic exposures,
the uncertainty induced by Global Climate Models (GCMs) and RCP scenarios also
plays an important role in climate impact assessment. Measured by the 90% range of
the changing characteristics of SPEI-3 from historical to 1.5°C warming world and from
1.5°C to 2.0°C warming target, the uncertainty induced by multi-model ensembles are
quantified in each grid under three RCPs (Figs. S5-6). Compared with the ensemble
mean change of SPEI-3 shown in Figs. 2-3, we find that the model uncertainty is
relatively large, particular for South America and Africa where the 90% range even
exceeds the ensemble mean change. This finding also holds true when evaluating the
drought duration and severity (Figs. S7-8), suggesting that model uncertainty cannot be
ignored in climate impact studies.
To fully consider model uncertainty on drought conditions, we also present the
bivariate return period of the present 50-year drought condition for each model under
RCP 4.5 in a 1.5°C warming world, and the occurrence change under an additional
0.5°C warming (Figs. S9-10). As expected, different climate models show large
variations, and several models even exhibit opposite changes over certain regions.
Despite this uncertainty, most models still project general increasing risks at the global




scale under climate warming, particularly for middle-latitude areas and tropics. For
RCP uncertainty, although we notice that the changing pattern of drought conditions
under three RCPs are similar to some extent (Figs. S5-8), we observe some differences
among RCP 2.6, RCP4.5 and RCP8.5 scenarios in several regions (especially in extra-
tropics). Given that these disparities deriving from different time periods due to the
warming level definition, they cannot perfectly represent the uncertainty of
concentration pathways. Despite this, they can still reflect that RCP uncertainty also
plays a role in climate impact studies, albeit model uncertainty usually accounts for a
dominated part.
Several previous studies (Wang et al., 2018; Gu et al., 2019; Chen et al., 2019) have
been devoted to detecting and attributing uncertainty to GCM structure, RCPs, internal
climate variability, and even drought indices and so on. Here, it is challenging to
consider all these uncertainties systematically; future work could focus on including the
integrated uncertainty and quantifying relative contributions on drought evolution and
impact assessments.

## 16 5. Conclusions

Motivated by the 2015 Paris Agreement proposal, we quantify the changes in global
drought bivariate magnitudes and socioeconomic consequences in the 1.5°C and 2.0°C
warmer worlds, with climate projected by the multi-model ensemble under three
representative concentration pathways (RCP2.6, 4.5, and 8.5). The drought
characteristics are identified using the SPEI combined with the run theory, and the
changes in occurrence are measured by both drought duration and severity, with the
incorporation of the copula functions and most likely realization method. The main
conclusions are summarized as follows (Table S1):
(1) The mean of SPEI-3 from the historical period to the 1.5°C and 2.0°C warmer
worlds are projected to descend at a global scale, while the standard deviation exhibits
large increases. As the SPEI-3 following the normal distribution, these changes suggest
that the distribution of SPEI-3 would shift towards the negative side with a flatter



tendency, implying a more severe drying condition in a future warming world.
(2) The drought duration is projected to slowly prolong across 78% of the land
surface, while the drought severity shows a much more pronounced rise globally in the
1.5°C warming world. Compared to 1.5°C warming condition, there will be a further
increase in drought severity and a persistent lengthening in drought duration under the
additional 2.0°C warming level. Several regions in middle-latitude regions and the
tropics would experience substantial increases in drought magnitude, such as Southeast
Asia, the Mediterranean, Southern Africa, Southern North America, and South America.
(3) More than 58% of global landmasses would be subject to twice more frequent
historical 50-year droughts even under the ambitious 1.5°C mitigation target. The
drought condition will further worsen under 2.0°C warming climate, with around a 9%
increase of the world landmasses experiencing such severe deterioration comparing to
the 1.5°C warming level.
(4) More than 75 (73) countries are projected to exhibit a 100% fraction in the
population (GDP) exposed to increasing drought risks even under the ambitious 1.5°C
warming trajectories. An extra 0.5°C warming will lead to an additional 17 countries
exhibiting a 100% fraction in socioeconomic exposure. Moreover, tropical countries
(i.e., Mexico and Brazil) will be subject to dramatically increased drought risks, with
85% of the land fraction would experiencing a doubled frequency of severe historical
droughts under the 1.5°C warming target; when the warming is increasing to 2.0°C, the
corresponding land fraction is projected to approach 90%.
**Data availability**
The climate simulation data can be accessed from the CMIP5 archive (https://esgf-
node.llnl.gov/projects/esgf-llnl/). The SSP data are provided by Prof. Buda Su and Prof.
Tong Jiang in National Climate Center, China Meteorological Administration.



## 1 Author contributions

JC conceived the original idea, and LG designed the methodology. JC, LPZ and JSK collected the data. LG developed the code and performed the study, with some contributions from JC and HMW. LG, JC, JBY, SCS and SLG contributed to the interpretation of results. LG and JBY wrote the paper, and JC, SCS, SLG, LPZ and JSK revised the paper.

## 8 Conflict of interest

The authors declare that they have no conflict of interest with the work presented here.

## 11 Acknowledgements

This work was partially supported by the National Key Research and Development Program of China (No. 2017YFA0603704; 2016YFC0402206), the National Natural Science Foundation of China (Grant Nos. 51779176, 51539009, 51811540407), the Overseas Expertise Introduction Project for Discipline Innovation (111 Project) funded by Ministry of Education and State Administration of Foreign Experts Affairs P.R. China (Grant No. B18037), and the Thousand Youth Talents Plan from the Organization Department of CCP Central Committee (Wuhan University, China). The authors would like to thank the World Climate Research Program working group on Coupled Modelling, and all climate modeling institutions listed in Table 1 for making GCM outputs available. We also thank Prof. Buda Su and Prof. Tong Jiang in National Climate Center, China Meteorological Administration for sharing the SSP data.

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

Managing the Risks of Extreme Events and Disasters to Advance Climate Change Adaptation.
A Special Report of Working Groups I and II of the Intergovernmental Panel on Climate
Change (IPCC), eds Field CB, et al. (Cambridge Univ Press, Cambridge, UK), 231-290, 2012.
Held, I. M. and Soden, B. J.: Robust responses of the hydrological cycle to global warming. J.
Climate, 19(21), 5686-5699, 2006.
Hintze, J. L. and Nelson, R. D.: Violin plots: a box plot-density trace synergism. The American
Statistician, 52(2), 181-184, 1998.
Hirabayashi, Y., Mahendran, R., Koirala, S., Konoshima, L., Yamazaki, D., Watanabe, S., and Kanae,
S.: Global flood risk under climate change. Nat. Clim. Change, 3(9), 816, 2013.
Huang, J., Yu, H., Dai, A., Wei, Y., and Kang, L.: Drylands face potential threat under 2 °C global
warming target. Nat. Clim. Change, 7(6), 417, 2017.
Huang, J., Qin, D., Jiang, T., Wang, Y., Feng, Z., Zhai, J., and Su, B.: Effect of Fertility Policy
Changes on the Population Structure and Economy of China: From the Perspective of the
Shared Socioeconomic Pathways. Earth's Future, 7(3), 250-265, 2019.
Hulme, M.: 1.5 °C and climate research after the Paris Agreement. Nat. Clim. Change, 6, 222, 2016.
Intergovernmental Panel on Climate Change (IPCC), 2018. Special Report on Global Warming of
1.5°C.
Jiang, T., Zhao, J., Jing, C., Cao, L. G., Wang, Y. J., Sun, H. M., and Wang, R.: National and
provincial population projected to 2100 under the shared socioeconomic pathways in China.



1       Clim. Chang. Res, 13, 128-137, 2017.

Jiang, T., Zhao, J., Cao, L., Wang, Y., Su, B., Jing, C., and Gao, C.: Projection of national and
3       provincial economy under the shared socioeconomic pathways in China. Advances in Climate
4       Change Research, 14(1), 50-58, 2018.

Jiao, Y., & Yuan, X. More severe hydrological drought events emerge at different warming levels
6       over the Wudinghe watershed in northern China. Hydrology and Earth System Sciences, 23(1),
7       621-635, 2019.

Jones, B. and O'Neill, B. C.: Spatially explicit global population scenarios consistent with the
9       Shared Socioeconomic Pathways. Environ. Res. Lett., 11(8), 084003, 2016.

Lehner, F., Coats, S., Stocker, T. F., Pendergrass, A. G., Sanderson, B. M., Raible, C. C., and
Smerdon, J. E.: Projected drought risk in 1.5°C and 2°C warmer climates. Geophys. Res. Lett.,
12      44: 7419-7428, 2017.

Leimbach, M., Kriegler, E., Roming, N., and Schwanitz, J.: Future growth patterns of world regions-
A GDP scenario approach. Glob. Environ. Change, 42:215-225, 2017.
Li, T., Guo, S., Liu, Z., Xiong, L., and Yin, J.: Bivariate design flood quantile selection using copulas.
Hydrol. Res., 48(4):997-1013, 2016.
Liu, K., & Jiang, D. Analysis of dryness/wetness over China using standardized precipitation
evapotranspiration index based on two evapotranspiration algorithms. Chinese Journal of
Atmospheric Sciences (in Chinese), 39(1), 23-36, 2015.
Liu, J., Wang, B., Cane, M. A., Yim, S. Y., and Lee, J. Y.: Divergent global precipitation changes
induced by natural versus anthropogenic forcing. Nature, 493(7434), 656-659.
https://doi.org/10.1038/nature11784, 2013.
Liu, W., Sun, F., Lim, W. H., Zhang, J., Wang, H., Shiogama, H., and Zhang, Y.: Global drought and
severe drought-affected populations in 1.5 and 2 °C warmer worlds. Earth Syst. Dynam., 9:267-
25      283, 2018.

Liu, X. F., Wang, S. X., Zhou, Y., Wang, F. T., Li, W. J., and Liu, W.L.: Regionalization and
spatiotemporal variation of drought in china based on standardized precipitation
evapotranspiration index (1961-2013). Adv. Meteorol., 18, 2015.
Lu, J., Vecchi, G. A., and Reichler, T.: Expansion of the Hadley cell under global warming. Geophys.
Res. Lett., 34, L06805. https:// doi.org/10.1029/2006GL028443, 2007.
Mishra, A. K. and Singh, V. P.: A review of drought concepts. J. Hydrol., 391(1-2), 202-216, 2010.
Mitchell, D., James, R., Forster, P. M., Betts, R. A., Shiogama, H., and Allen, M.: Realizing the
impacts of a 1.5 C warmer world. Nat. Climat. Change, 6(8), 735, 2016.
Muller, C. J., O'Gorman, P. A., and Back, L. E.: Intensification of precipitation extremes with
warming in a cloud-resolving model. J. Clim., 24(11), 2784-2800, 2011.
Naumann, G., Alfieri, L., Wyser, K., Mentaschi, L., Betts, R. A., Carrao, H., and Feyen, L.: Global
changes in drought conditions under different levels of warming. Geophys. Res. Lett., 45(7),
38      3285-3296, 2018.

O'Neill, B. C., Kriegler, E., Riahi, K., Ebi, K. L., Hallegatte, S., Carter, T. R., and van Vuuren, D.
P.: A new scenario framework for climate change research: the concept of shared
socioeconomic pathways. Climatic change, 122(3), 387-400, 2014.
Park, C. E., Jeong, S. J., Joshi, M., Osborn, T. J., Ho, C. H., Piao, S., and Kim, B. M.: Keeping
global warming within 1.5° C constrains emergence of aridification. Nat. Clim. Change, 8(1),
44      70, 2018.

Peters, G. P.: The best available science to inform 1.5 C policy choices. Nat. Clim. Change, 6(7),
646. https://doi.org/10.1038/nclimate3000, 2016.
Routson, C. C., C. A. Woodhouse, J. T. Overpeck, J. L. Betancourt, J. L., and McKay, N.P.:
Teleconnected ocean forcing of Western North American droughts and pluvials during the last
millennium, Quat. Sci. Rev., 146, 238-250, 2016.
Salvadori, G., De Michele, C., and Durante, F. Multivariate design via copulas. Hydrol. Earth Syst.
Sc. 8, 5523–5558, 2011.
Samir, K. C. and Lutz, W.: The human core of the shared socioeconomic pathways: Population
scenarios by age, sex and level of education for all countries to 2100. Global Environmental
Change, 42, 181-192, 2017.
Sanderson, B. M., Xu, Y., Tebaldi, C., et al.: Community climate simulations to assess avoided
impacts in 1.5 and 2 °C futures. Earth Syst. Dynam., 8, 827-847, https://doi.org/10.5194/esd-



1    8-827-2017, 2017.
Scheff, J. and Frierson, D. M.: Scaling potential evapotranspiration with greenhouse warming. J.
3       Clim., 27(4), 1539-1558, 2014.
Seager, R., Y. Kushnir, C. Herweijer, N. Naik, and J. Velez.: Modeling of tropical forcing of
5       persistent droughts and pluvials over western North America: 1856–2000, J. Clim., 18, 4065-
6       4088, doi:10.1175/JCLI3522.1, 2005.
Schilling, J., Freier, K.P., Hertig, E. and Scheffran, J.: Climate change, vulnerability and adaptation
8       in North Africa with focus on Morocco. Agric. Ecosyst. Environ., 156, 12-26, 2012.
Smirnov, O., Zhang, M., Xiao, T., Orbell, J., Lobben, A., and Gordon, J.: The relative importance of
climate change and population growth for exposure to future extreme droughts. Climatic
Change, 138(1-2), 41-53, 2016.
Su, B., Huang, J., Fischer, T., Wang, Y., Kundzewicz, Z. W., Zhai, J., and Tao, H.: Drought losses in
China might double between the 1.5° C and 2.0° C warming. P. Natl. Acad. Sci. USA., 115(42),
14      10600-10605, 2018.
Touma, D., Ashfaq, M., Nayak, M.A., Kao, S.-C., Diffenbaugh, N.S.: A multi-model and multi-
index evaluation of drought characteristics in the 21st century. J. Hydrol., 526, 196–207.
http://dx.doi.org/10.1016/j.jhydrol.2014.12.011, 2015.
Tsakiris, G., Kordalis, N., Tigkas, D., Tsakiris, V., and Vangelis, H.: Analysing drought severity and
areal extent by 2D Archimedean copulas. Water Resour. Manage., 30, 1-13, 2016.
UNFCCC, 2015. Conference of the Parties. Adoption of the Paris Agreement, Paris. 1-32.
Vautard, R., Gobiet, A., Sobolowski, S., Kjellström, E., Stegehuis, A., Watkiss, P. and Jacob, D.:
The European climate under a 2 C global warming. Environ. Res. Lett., 9(3), 034006, 2014.
Vicente-Serrano, S.M., Beguería, S., and López-Moreno, J.I.: A Multiscalar Drought Index Sensitive
to Global Warming: The Standardized Precipitation Evapotranspiration Index. J. Clim., 23(7):
1696-1718. DOI:10.1175/2009jcli2909.1, 2010.
Wang, H. M., Chen, J., Cannon, A. J., Xu, C. Y. and Chen, H.: Transferability of climate simulation
uncertainty to hydrological impacts. Hydrol. Earth Syst. Sc., 22(7), 3739-3759, 2018.
Wen, S. S., Wang, A. Q., Tao, H., Malik, K., Huang, J., Zhai, J., Jing, C., Rasul, G. and Su B.:
Population exposed to drought under the 1.5 °C and 2.0 °C warming in the Indus River Basin.
Atmos. Res., 218: 296-305, 2019.
Wong, G., Van Lanen, H.A.J. and Torfs, P.J.J.F.: Probabilistic analysis of hydrological drought
characteristics using meteorological drought. Hydrol. Sci. J., 58 (2), 253-270, 2013.
Xie, S. P., Deser, C., Vecchi, G. A., Ma, J., Teng, H. and Wittenberg, A.: Global warming pattern
formation: Sea surface temperature and rainfall. J. Climate, 23(4), 966–986.
https://doi.org/10.1175/2009JCLI3329.1, 2010.
Xiao, M., Zhang, Q., Singh, V. P., & Chen, X. Probabilistic forecasting of seasonal drought
behaviors in the Huai River basin, China. Theoretical and applied climatology, 128(3-4), 667-
38      677, 2017.
Xu, K., Yang, D. W., Xu, X. Y., and Lei, H. M.: Copula based drought frequency analysis
considering the spatio-temporal variability in Southwest China. J. Hydrol. 527, 630-640, 2015.
Yin, J. B., Guo, S. L., He, S. K., Guo, J. L., Hong, X. J., and Liu, Z. J.: A copula-based analysis of
projected climate changes to bivariate flood quantiles. J. Hydrol. 566, 23-42, 2018a.
Yin, J. B., Gentine, P., Zhou, S., Sullivan, C. S., Wang, R., Zhang, Y., and Guo, S.L.: Large increase
in global storm runoff extremes driven by climate and anthropogenic changes. Nat. Commun.
9, 4389, 2018b.
Yin, J. B., Guo, S., Wu, X., Yang, G., Xiong, F., and Zhou, Y.: A meta-heuristic approach for
multivariate design flood quantile estimation incorporating historical information. Hydrol. Res.,
48      50(2), 526-544, 2019.
Yevjevich, V. M.: Objective approach to definitions and investigations of continental hydrologic
droughts, An. Hydrology papers (Colorado State University); 23., 1967.
Yu, M., Li, Q., Hayes, M. J., Svoboda, M. D., and Heim, R. R.: Are droughts becoming more
frequent or severe in China based on the standardized precipitation evapotranspiration index:
1951-2010?. Int. J. Climatol.,34(3), 545-558, 2014.
Yuan, X., Zhang, M., Wang, L., & Zhou, T. Understanding and seasonal forecasting of hydrological
drought in the Anthropocene. Hydrology and Earth System Sciences, 21(11), 5477-5492, 2017.
Zargar, A., Sadiq, R., Naser, B., and Khan, F. I.: A review of drought indices. Environ. Reviews, 19,





1       333-349, 2011.
2   Zhang, Q., Xiao, M.Z., and Singh, V.P.: Uncertainty evaluation of copula analysis of hydrological
3       droughts in the East River basin, China. Global Planet., Change, 129, 1-9, 2015.
4   Zscheischler, J. and Seneviratne, S. I.: Dependence of drivers affects risks associated with
5       compound events. Sci. Adv., 3(6), e1700263, 2017.





1   **List of Tables**





1    **Table 1 Information about the 13 GCMs used in this study**

| No. | Model name | Resolution | Institution |
|---|---|---|---|
| 1 | BNU-ESM | 2.8 × 2.8 | Collegeof Global Change and Earth System Science, Beijing Normal University |
| 2 | CanESM2 | 2.8 × 2.8 | Canadian Centre for Climate Modelling and Analysis |
| 3 | CNRM-CM5 | 1.4 × 1.4 | Centre National de Recherches Météorologiques and Centre Européen de Recherche et Formation Avancée en Calcul Scientifique |
| 4 | CSIRO-Mk3.6.0 | 1.8 ×1.8 | Commonwealth Scientific and Industrial Research Organization and Queensland Climate Change Centre of Excellence |
| 5 | GFDL-CM3 | 2.5 × 2.0 | NOAA Geophysical Fluid Dynamics Laboratory |
| 6 | GFDL-ESM2G | 2.5 × 2.0 | |
| 7 | GFDL-ESM2M | 2.5 × 2.0 | |
| 8 | IPSL-CM5A-LR | 3.75 × 1.9 | Institut Pierre Simon Laplace |
| 9 | IPSL-CM5A-MR | 2.5 × 1.25 | |
| 10 | MIROC-ESM-CHEM | 2.8 × 2.8 | Japan Agency for Marine-Earth Science and Technology, Atmosphere and Ocean Research Institute (The University of Tokyo), and National Institute for Environmental Studies |
| 11 | MIROC-ESM | 2.8 × 2.8 | |
| 12 | MIROC5 | 1.4 × 1.4 | Atmosphere and Ocean Research Institute (The University of Tokyo), National Institute for Environmental Studies, and Japan Agency for Marine-Earth Science and Technology |
| 13 | MRI-CGCM3 | 1.1 × 1.1 | Meteorological Research Institute |



1  **Table 2 Drought Categories in the SPEI**

| SPEI | Categories |
|---|---|
| >-0.5 | Near Normal |
| -1.0 to -0.5 | Mild drought |
| -2.0 to -1.0 | Moderate drought |
| <-2.0 | Extremely drought |



4 **List of Figures**



32    Fig. 15. Projected changes in joint 50-year return periods of droughts when using the -

33    0.5 as the threshold and the -0.8 as the threshold between the 1.5°C and 2.0°C warming

34    target

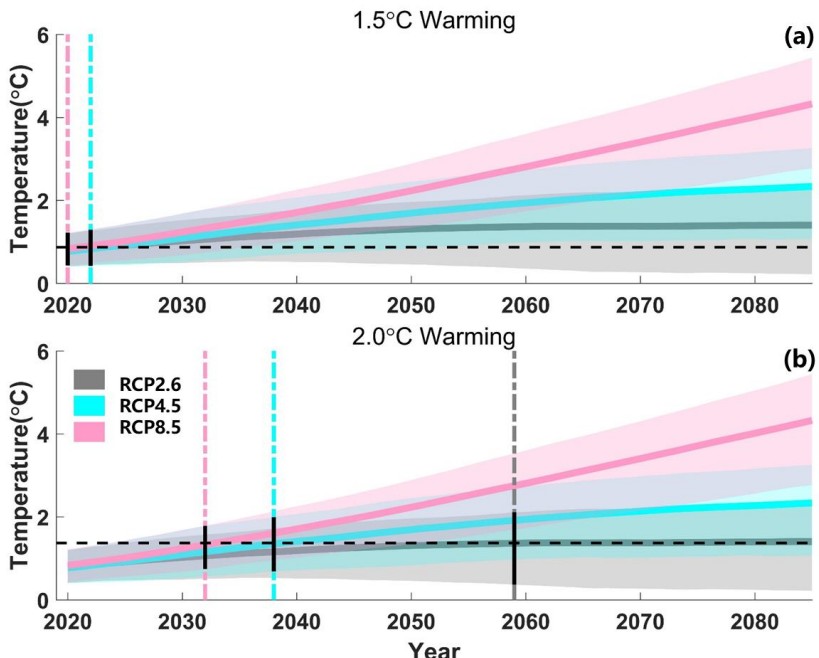

**Fig. 1. Projected global mean temperatures when reaching 1.5°C warming (**a**) and**

**2.0°C warming (**b**).**

Development of centered 30-year global average temperatures for all 13 General

Circulation Models (GCMs) and 3 Representative Concentration Pathways (RCPs)

included in this study. The vertical dark lines mark the uncertainty when the warming

target is reached. In **Fig.1a**, the determined time in RCP26 is the same with that in

RCP45, so the vertical dashed grey line is covered by the dashed cyan line.





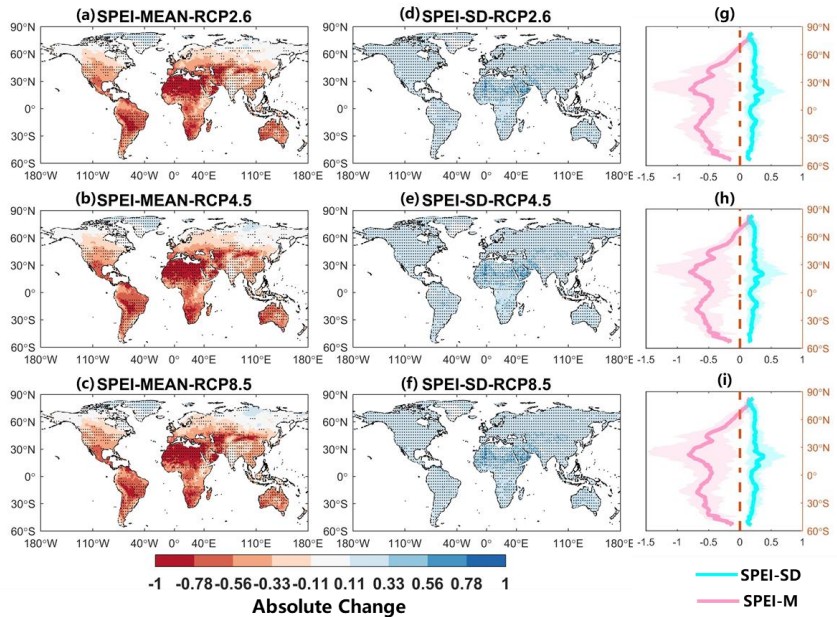

**Fig. 2. Projected changes in the mean and standard deviation of SPEI under the**

**1.5°C warming target**

Maps of the projected changes in the mean (**a,c,e**) and standard deviation (**b,d,f**) of

SPEI from historical reference period (1976-2005) to the 1.5°C warming target under

RCP2.6, RCP4.5, and RCP8.5. (**g,h,i**) Zonal results for changes in 1° latitude bin.

The stippling (**a-f**) is shaded for areas where at least 80% (i.e., 10 out of 13) of the

GCMs agree on the sign of the change.



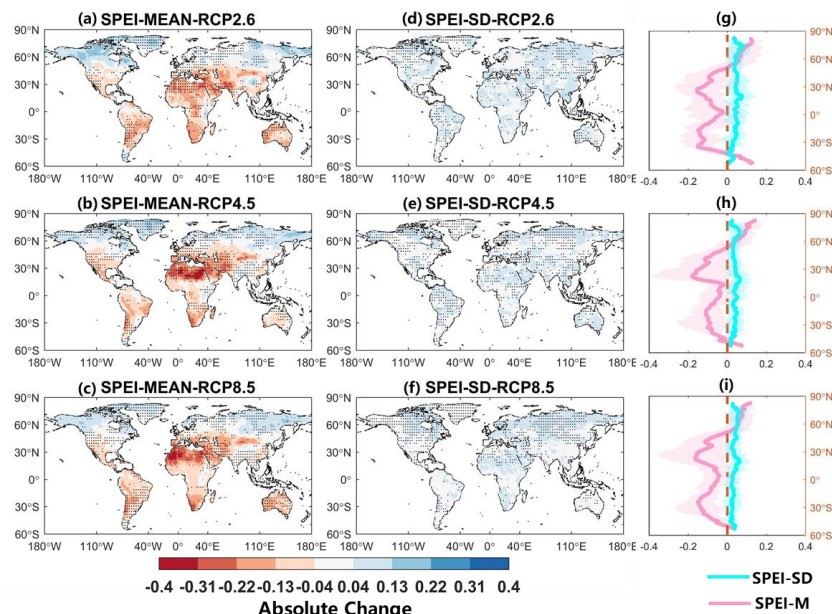

52

**Fig. 3. Projected changes in the mean and standard deviation of SPEI between the**

**1.5°C and 2.0°C warming target**

Maps of the projected changes in the mean (**a,c,e**) and standard deviation (**b,d,f**) of

SPEI from 1.5°C to the 2.0°C warming target under RCP2.6, RCP4.5, and RCP8.5.

(**g,h,i**) Zonal results for changes in 1° latitude bin. The stippling (**a-f**) is shaded for areas

where at least 80% (i.e., 10 out of 13) of the GCMs agree on the sign of the change.





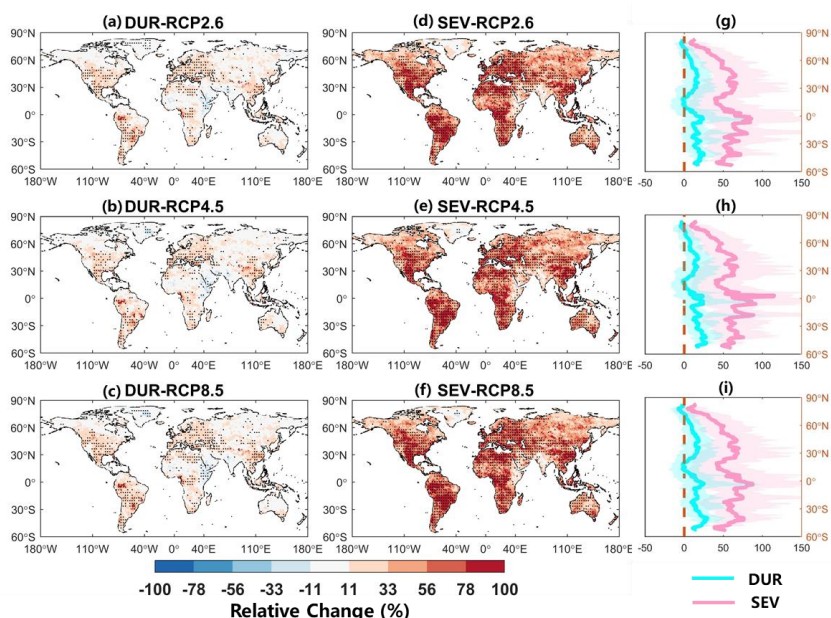


**Fig. 4. Projected changes in drought duration and severity under the 1.5°C warming target**

Maps of the relative changes (%) in the multi-model ensemble mean drought duration (**a,c,e**) and drought severity (**b,d,f**) from the reference period (1976-2005) to the 1.5°C warming target under RCP2.6, RCP4.5, and RCP8.5. (**g,h,i**) Zonal results for drought duration and severity in 1° latitude bin. The stippling (**a-f**) is shaded for areas where at least 80% (i.e., 10 out of 13) of the GCMs agree on the sign of the change.

70




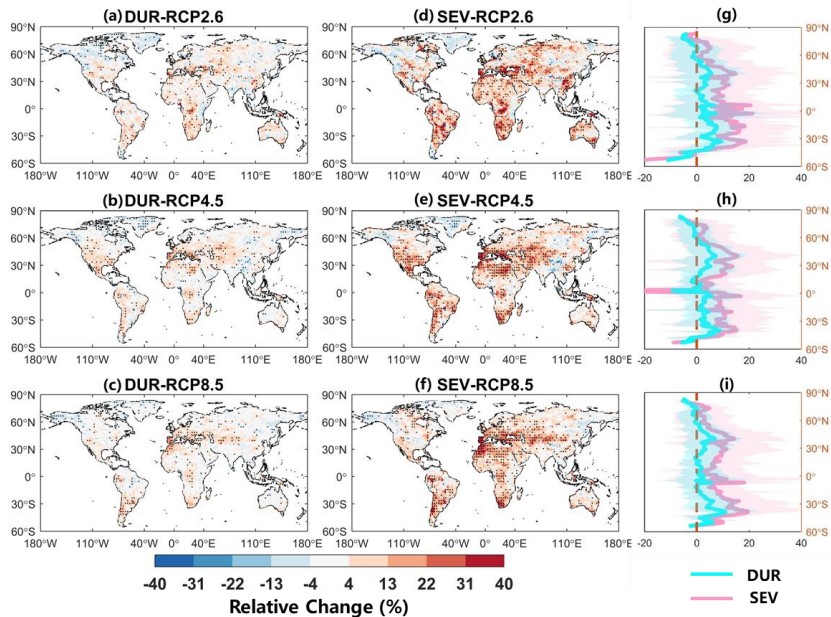

**Fig. 5. Projected changes in drought duration and severity between the 1.5°C and 2.0°C warming target**

Maps of the relative changes (%) in the multi-model ensemble mean drought duration (**a,c,e**) and drought severity (**b,d,f**) from the 1.5°C to the 2.0°C warming target under RCP2.6, RCP4.5, and RCP8.5. (**g,h,i**) Zonal results for drought duration and severity in 1° latitude bin. The stippling (**a-f**) is shaded for areas where at least 80% (i.e., 10 out of 13) of the GCMs agree on the sign of the change.



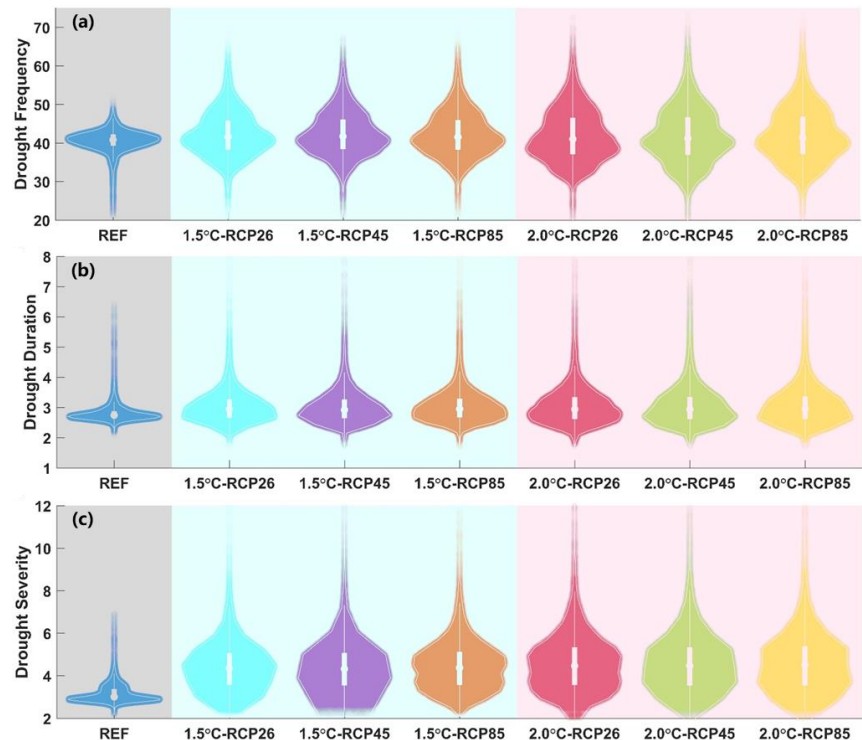

80

**Fig. 6. Distributions for drought characteristics under different time periods**

Distributions in the multi-model ensemble mean drought frequency (**a**), drought
duration (**b**) in months, and drought severity (**c**) across global land areas for the
reference period (1976-2005), the 1.5°C, and the 2.0°C warming target, respectively.

85

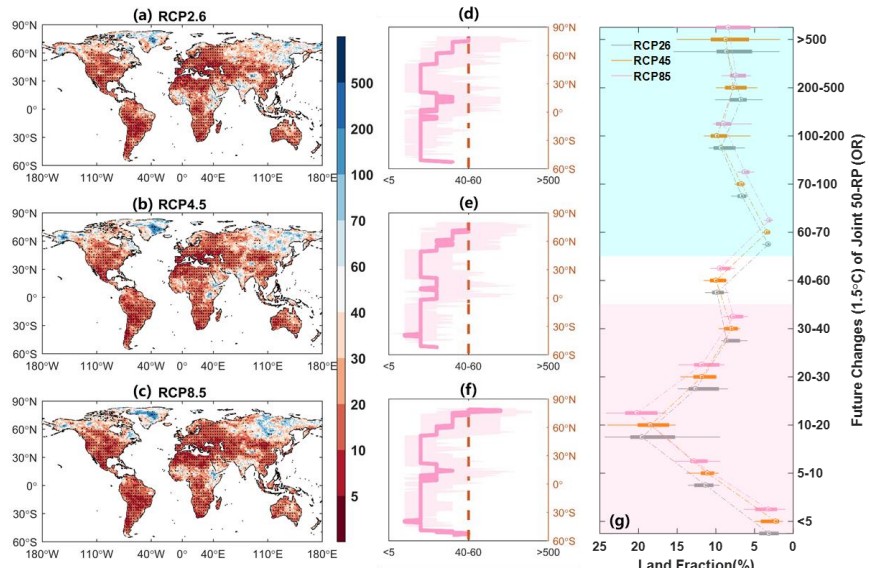

**Fig. 7. Projected changes in joint 50-year return periods of droughts under the**

**1.5°C warming target**

Projected GCMs median changes in joint 50-year return periods of droughts (duration

and severity) from the reference period to the 1.5°C warming target under RCP2.6,

RCP4.5, and RCP8.5. (**d,e,f**) Zonal results in each 1° latitude bin; (**g**) Global land

fraction for each change category. The stippling (**a-c**) is shaded for areas where at least

80% (i.e., 10 out of 13) of the GCMs agree on the sign of the change.

93

94





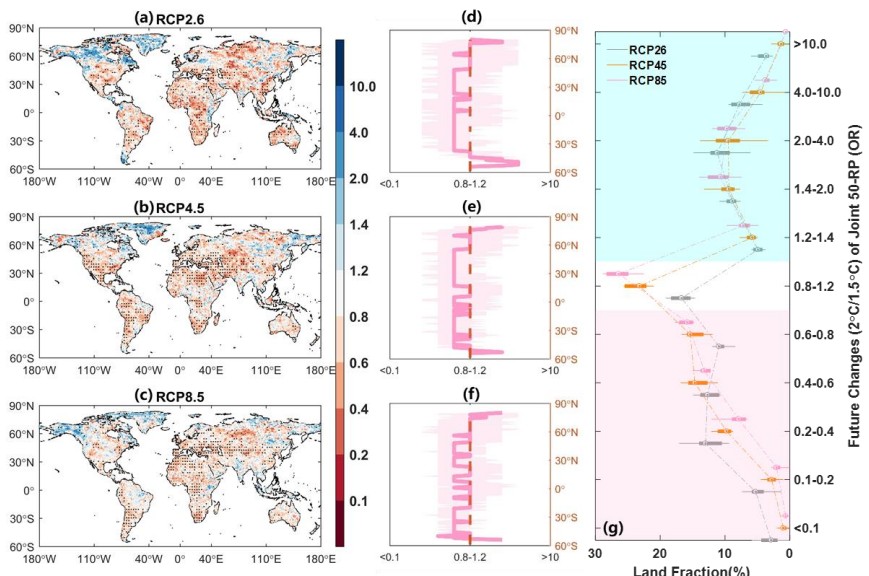

95

**Fig. 8. Projected changes in joint 50-year return periods of droughts between the**

**1.5°C and 2.0°C warming target**

Projected GCMs median changes in joint 50-year return periods of droughts (duration

and severity) from the 1.5°C to the 2.0°C warming target under RCP2.6, RCP4.5, and

RCP8.5. (**d,e,f**) Zonal results in each 1° latitude bin; (**g**) Global land fraction for each

change category. The stippling (**a-c**) is shaded for areas where at least 80% (i.e., 10 out

of 13) of the GCMs agree on the sign of the change.




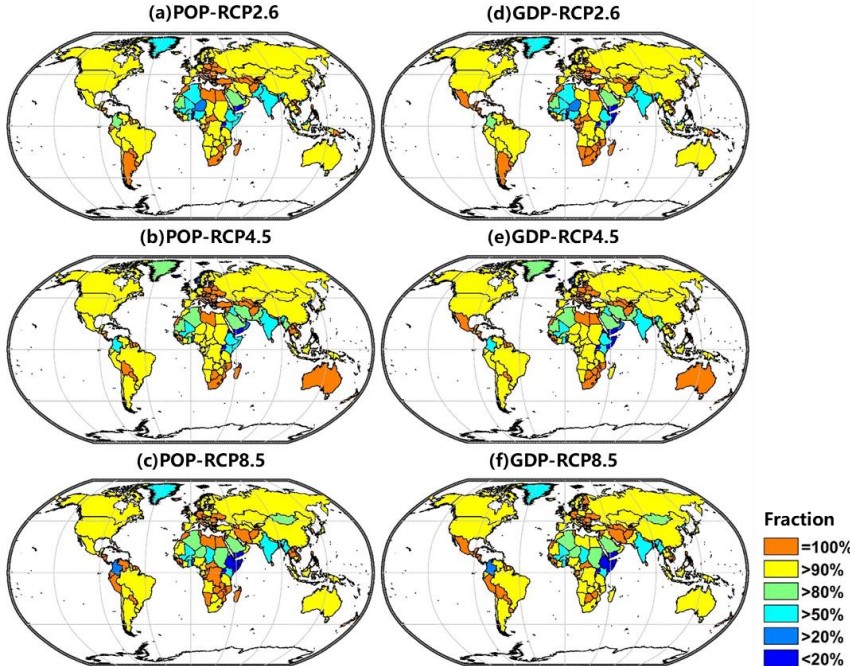


**Fig. 9. National population and GDP fraction exposing to more frequent severe droughts under the 1.5°C warming target**

Maps of the population (**a,c,e**) and Gross Domestic Product (GDP) (**b,d,f**) fractions that exposed to increasing drought risks from the reference period to the 1.5°C warming target under RCP2.6, RCP4.5, and RCP8.5 scenarios. The color-bar in the right side represents six ranks of the population and GDP fractions.

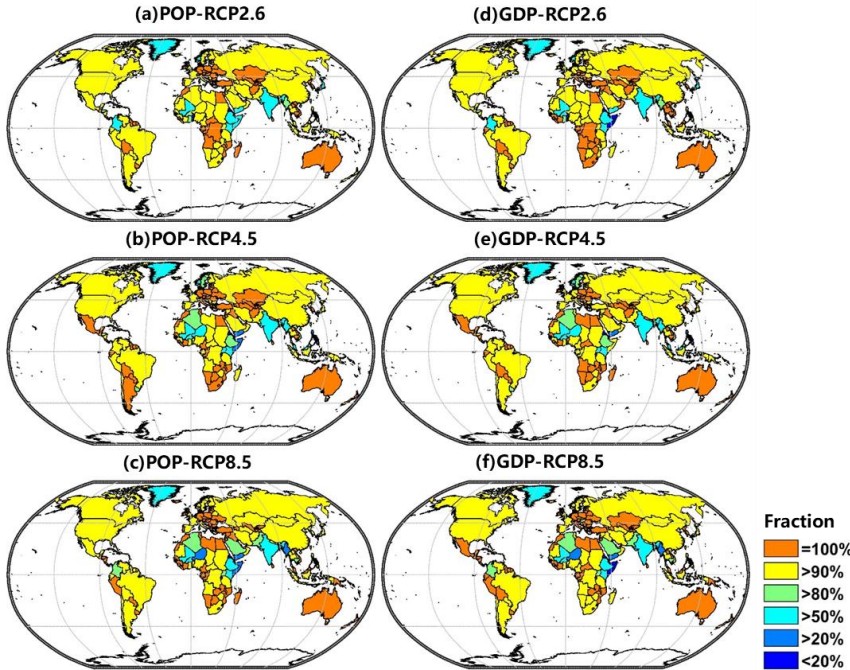


**Fig. 10. National population and GDP fraction exposing to more frequent severe**

**droughts under the 2.0°C warming target**

Maps of the population (**a,c,e**) and Gross Domestic Product (GDP) (**b,d,f**) fractions that
exposed to increasing drought risks from the reference period to the 2.0°C warming
target under RCP2.6, RCP4.5, and RCP8.5 scenarios. The color-bar in the right side
represents six ranks of the population and GDP fractions.



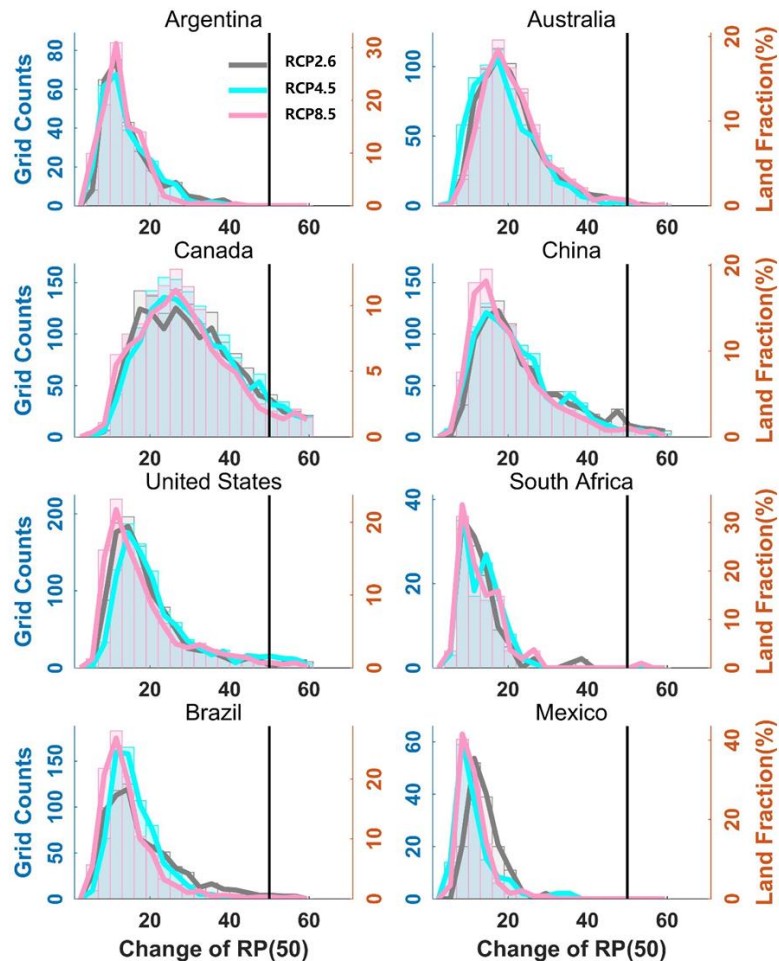


**Fig. 11. Projected changes of drought risks for 8 typical drought-prone countries**

**under the 1.5°C warming target**

Projected GCMs median changes in joint 50-year return periods of droughts (duration

and severity) as a function of land fraction for 8 typical drought-prone countries from

the reference period to the 1.5°C warming target under RCP2.6, RCP4.5, and RCP8.5.



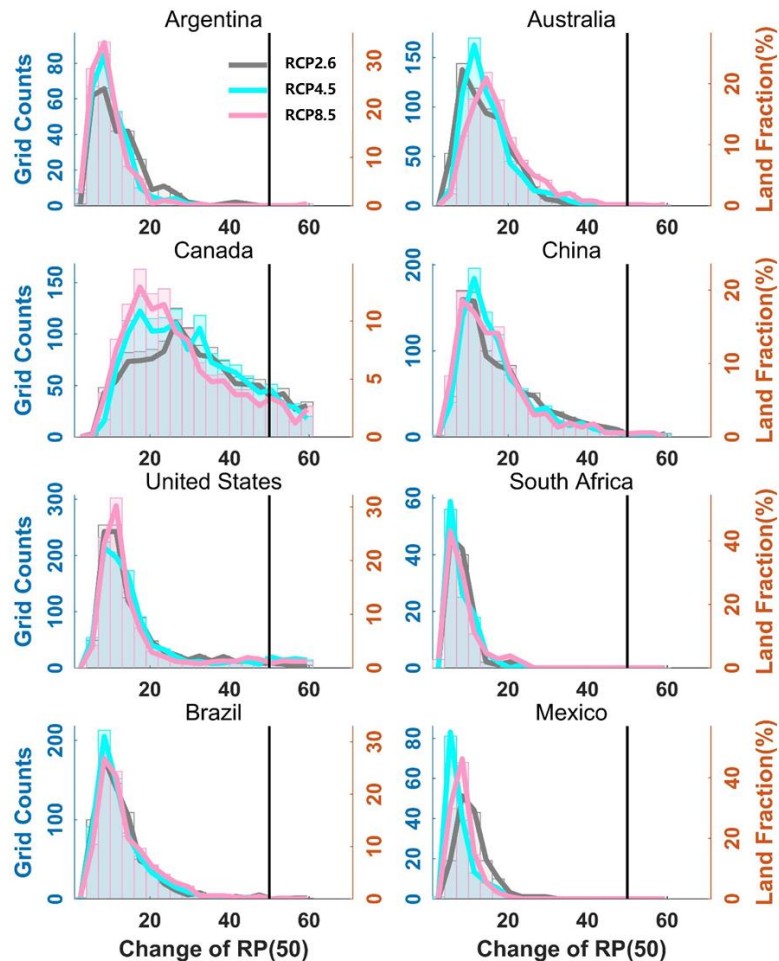

131

**Fig. 12. Projected changes of drought risks for 8 typical drought-prone countries**

**under the 2.0 °C warming target**

Projected GCMs median changes in joint 50-year return periods of droughts (duration

and severity) as a function of land fraction for 8 typical drought-prone countries from

the reference period to the 2.0°C warming target under RCP2.6, RCP4.5, and RCP8.5.





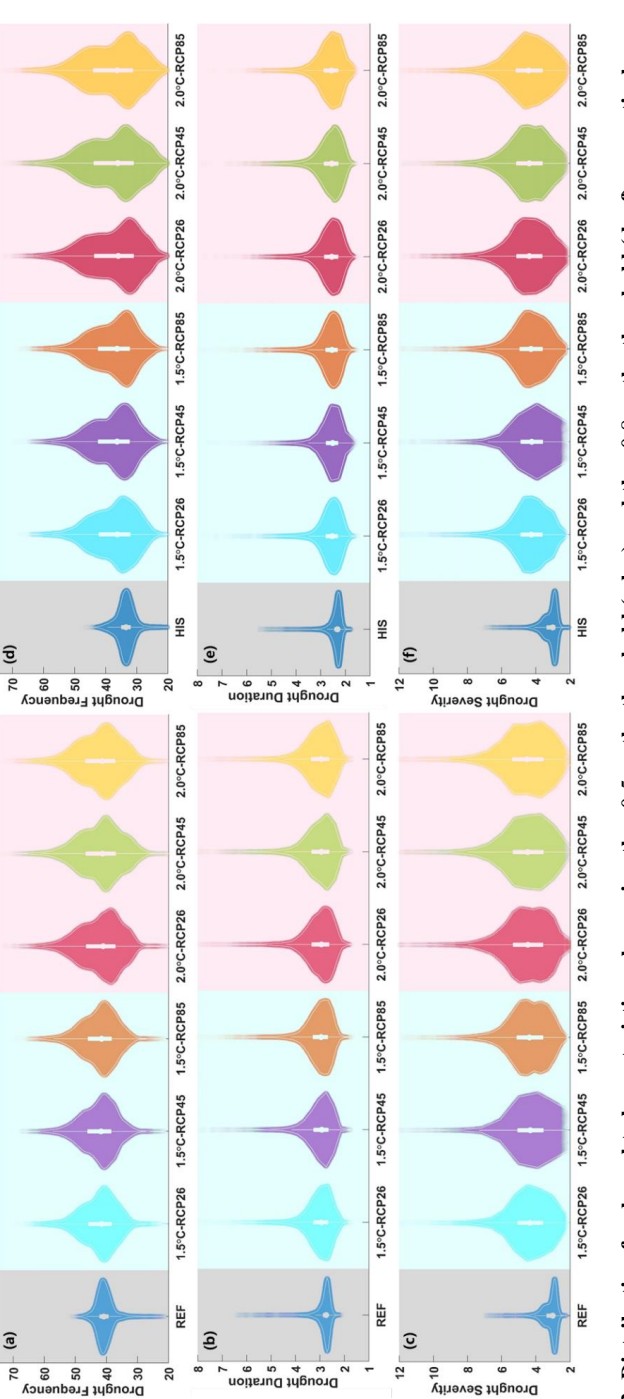

**Fig. 13. Distribution for drought characteristics when using the -0.5 as the threshold (a,b,c) and the -0.8 as the threshold (d,e,f), respectively.**





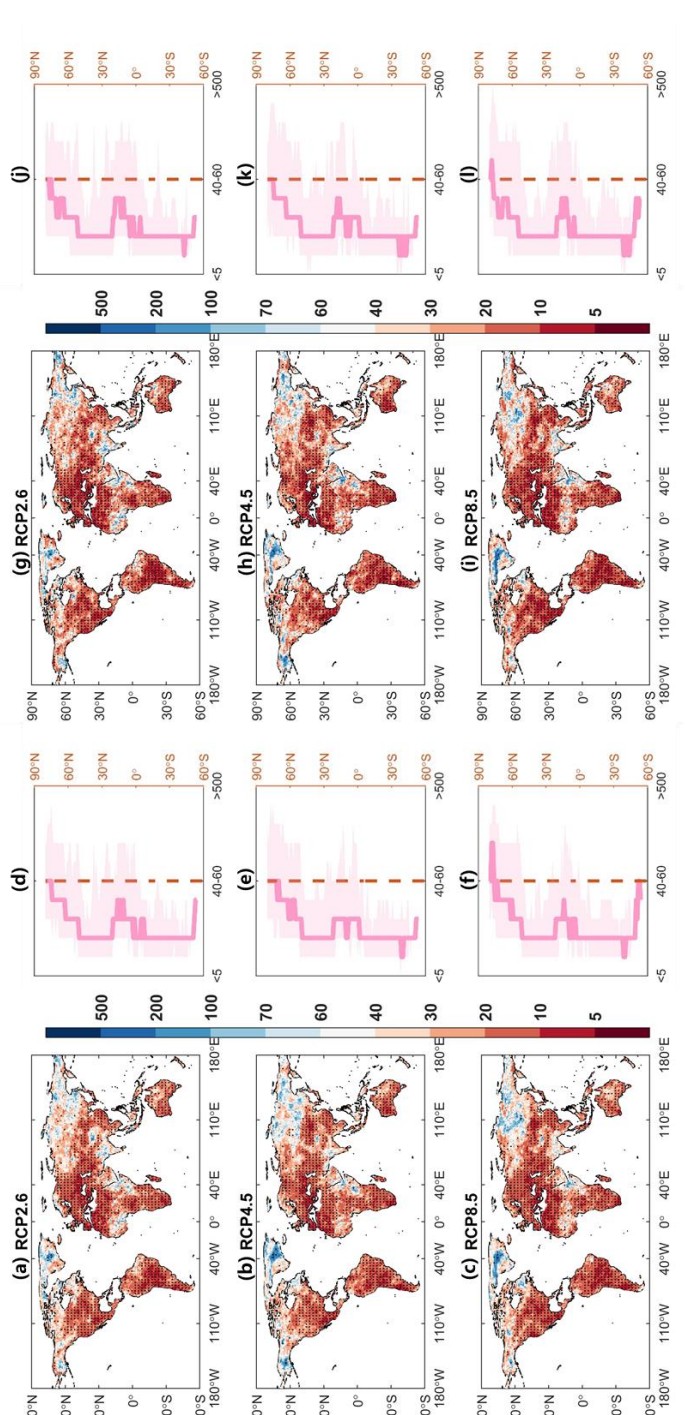

**Fig. 14 Projected changes in joint 50-year return periods of droughts when using the -0.5 as the threshold (a-f) and the -0.8 as the threshold (g-l) under**

**the 1.5°C warming target**






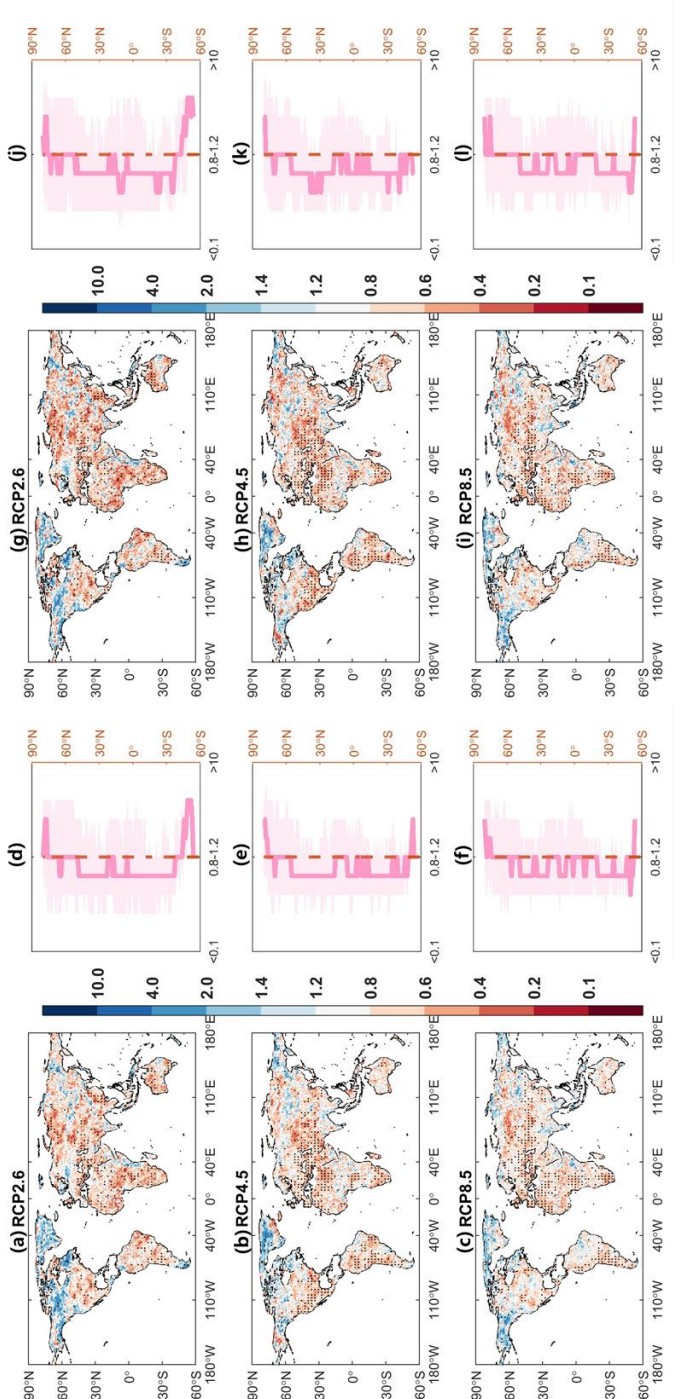


**Fig. 15 Projected changes in joint 50-year return periods of droughts when using the -0.5 as the threshold (a-f) and the -0.8 as the threshold (g-l)**


**between the 1.5°C and 2.0°C warming target**

