# Peer review of "Projected increases in magnitude and socioeconomic exposure of"

_Hydrology and Earth System Sciences, 2019_

## Referee Comment (RC1) · Anonymous Referee #1 · 30 Oct 2019

In this manuscript, the authors used the SPEI and run theory to define drought events, analyzed the variations of drought severity and duration by joint return period based on copula function and highlighted changes in exposures of population and GDP to global drought under three RCP scenarios (corresponding to three SSPs) at 1.5°C and 2 °C warming targets. The idea of studying the socioeconomic exposures to global drought is meaningful for countries concerned to understand and mitigate potential drought risks in the future. Generally, the manuscript is well organized with clear logic, before I recommend it for publication, major improvements are still needed.

When discussing the increase in the magnitude of global drought, the severity and duration of drought are both considered using a copula function and the drought is defined using SPEI< -0.5 and run theory, the methods are all good. As in table 2 indicates, SPEI <-0.5 incorporates three different levels of drought from mild, moderate to extreme drought. The authors used copula function to consider both the severity and duration, however, the severity of drought retrieved from the run theory may not reveal the distribution of different levels of drought? Although authors discussed the threshold of 0.8 to confirm relevant results, whether the selection of this threshold may further influence the results of socioeconomic exposures to droughts is worth thinking.

When calculating SPEI with Penman-Monteith-based PET, the term $(0.34u_2)$ in the equation is finally obtained through the ratio $r_s/r_a$ and represents the suggested reference crop surface (assuming a standard plant height of 0.12 m, affixed surface resistance of 70 $sm^{-1}$ and an albedo of 0.23). However, considering a distinct vegetation response to elevated $CO_2$ as simulated in the fully coupled climate models, it is important to point out that some of the assumptions that underlie the computation of PET (and thus SPEI) are incorrect (or at least the projected drought is not so severe) under conditions of changing $CO_2$ concentrations (Greve et al., 2019, ERL; Yang et al., 2018, NCC; Roderick et al., 2015, WRR).The authors should at least discuss the potential impacts of the elevated $CO_2$ on their drought risk assessment in Section 4.

Given the relative coarseness of the CMIP5 models, I think interpolation of the results (especially bilinearly interpolated P and PET to a common resultion before calculating SPEI with them) to 1 degree spatial resolution is not appropriate. A 2 degree commen grid would be better, and would avoid effectively making up data at the much finer resolution. The authors should at least discuss the impact of interpolation on their results in the maintext.

Some specific parts need further clarification.
1.  During the investigation regarding the exposures of population and GDP to droughts under three RCP scenarios at two warming levels, for example, under the RCP8.5 scenario (SSP5), the specific time when future warming reaches 1.5°C or 2°C

under RCP 8.5 can be different (from Fig 1), population and GDP can also possibly differ in two climates. From Line 17-Line 25 (Page 11), did the author suggest that the dynamic of population and GDP under RCP 8.5 at two warming climates was also considered using the multi-year average? If so, in section 3.4 about population and GDP exposure from increasing drought risks, it was concluded that a large percentage of population and GDP will be exposed to increasing drought risk. The drought risk has been increasing with warming climate, let's say if population and GDP have been increasing with time, then which one contributes to the increasing exposures, the increasing population or the increasing drought risks, I think this is a key question that authors should clarify when assessing the socioeconomic exposure.

2. Page 11, Line 13-16, how is the ratio of the recalculated recurrence frequency calculated and why a less than 1.0 ratio suggests worrisome drought condition. Need further clarification.

3. Page 12 section 3.1 projected changes in dryness, the author used SPEI and the run theory to define drought event, and the title of the manuscript is about the global drought, why would authors use SPEI to explain the dryness instead of using the defined event to study the changes in global drought for consistency.

4. Page 15 Line 28-29, whether the fraction of drought-affected population (or GDP) divided by total population (or GDP) can be a fairer and more impartial assessment is really hard to say given the fact that this method seems to cover up some most drought-affected countries, like the United States and China.

5. Generally, in the discussion of either the magnitude of drought or the socioeconomic exposures of droughts, the differences between two warming targets are highlighted, however, the differences among three RCP scenarios are barely discussed in the manuscript. It makes me doubt the reason and necessity of using three RCP scenarios since they present almost similar variations under two warming targets. This issue might be even obvious in Fig 9 and 10, for example, in Fig 9, under RCP 4.5, population and GDP suggest 100% exposure to drought in Australia, which drops to 90% under RCP 8.5. Possible reasons and texts are needed here.

6. Not sure whether section 3.5 is necessary since similar conclusions have been achieved in Fig 7 and 8, and these typical countries can just be used for further explanations in section 3.3. Besides, additional explanations for Fig 7g and Fig 8g are very necessary.

Minor suggestions.
1. Citation of Fig. 3 somewhere between lines 21 and 22 in Page 12.

2. Writing in the manuscript should be more concise in the data and method section, e.g. Page 6 line 7, use surface maximum, mean, minimum air

temperature to avoid repeat.

3. Table 2, extreme drought instead of extremely drought

---

## Referee Comment (RC2) · Anonymous Referee #2 · 24 Dec 2019

General Comments

In summary, motivated by the 2015 Paris Agreement proposal, this manuscript calculated global 3-month Standardized Precipitation Evapotranspiration Index (SPEI-3) based on 13 CMIP5 GCM simulations under three RCP scenarios (RCP2.6/4.5/8.5) during 1976-2100, quantified changes in global drought duration, severity and occurrence under a bivariate framework, and analyzed the drought exposures of populations and regional GDP under 3 shared socioeconomic pathways (SSPs) in future 1.5 and 2-degree warming worlds. Generally, this well written manuscript is able to portray drought evolution with different warming trajectories and provide information for climate

adaptation strategies. Here I list several questions below and suggest acceptance of the manuscript after minor revision.

Minor Comments

(1) P7, L19-20. How do you determine the year in which a specific warming target is achieved? I suppose you select the median year of the 30-year period with surface temperature closest to the warming target for each RCP (not for each RCP-GCM combination), so that the reaching year is the same for all 13 GCMs under a prescribed RCP scenario. The authors should clarify this.

(2) Figure 1a. I've noticed that the determined years under both scenarios (RCP2.6 and RCP8.5) are the first year (2020) of the whole period. Is it possible that for some GCM future projections, 1.5-degree warming (or even higher) has already been reached even at the beginning of the simulation period? If so, maybe it could affect the results in this manuscript.

(3) Figure 9, 10, S2 and Discussion Section. There are several countries (e.g. the United States) will experience a decrease in POP and GDP fraction exposing to more frequent severe droughts under the 2-degree warming level compared to 1.5-degree. I will be appreciated if the author could provide possible reasons considering the increasing drought risks in these countries.

(4) P10, L26. Eq. (5) should be Eq. (11)?

(5) As the authors mentioned in Section 2.1 that RCP2.6 is associated with SSP1, I suggest the author use SSP126 instead of RCP2.6 when talking about future drought risks. Same with SSP245 and SSP585.

(6) Relative to huge gaps in drought characteristics for two warming targets, results under three RCP scenarios seems to have few differences (e.g. Figure 6). Maybe the authors could explain the reason in Discussion Section, or explore the possible causes in future studies.

---

## Author Response (AR1)

Replies to Referee #1

**Projected increases in magnitude and socioeconomic exposure of global droughts in 1.5 °C and 2 °C warmer climates**

Lei Gu, Jie Chen, Jiabo Yin, Sylvia C. Sullivan, Hui-Min Wang, Shenglian Guo, Liping Zhang, and Jong-Suk Kim

We thank the anonymous referee for the constructive comments and queries. We have provided detailed responses to each comment below and will revise the manuscript accordingly. For clarity, comments are given in *italics*, and our responses are given in plain text.

Authors' responses

**Legend**

*Reviewer's comments*

Authors' responses

*In this manuscript, the authors used the SPEI and run theory to define drought events, analyzed the variations of drought severity and duration by joint return period based on copula function and highlighted changes in exposures of population and GDP to global drought under three RCP scenarios (corresponding to three SSPs) at 1.5ºC and 2ºC warming targets. The idea of studying the socioeconomic exposures to global drought is meaningful for countries concerned to understand and mitigate potential drought risks in the future. Generally, the manuscript is well organized with clear logic, before I recommend it for publication, major improvements are still needed.*

We appreciate the reviewer's positive evaluation and professional comments on our manuscript. Please find our point-by-point responses below.

*1. When discussing the increase in the magnitude of global drought, the severity and duration of drought are both considered using a copula function and the drought is defined using SPEI< -0.5 and run theory, the methods are all good. As in table 2 indicates, SPEI <-0.5 incorporates three different levels of drought from mild, moderate*

*to extreme drought. The authors used copula function to consider both the severity and duration, however, the severity of drought retrieved from the run theory may not reveal the distribution of different levels of drought? Although authors discussed the threshold of 0.8 to confirm relevant results, whether the selection of this threshold may further influence the results of socioeconomic exposures to droughts is worth thinking.*

Reply: Thanks for this comment. We agree with the reviewer that the selection of threshold needs to be further clarified. Please find it as follows:

In the run theory, once the threshold (e.g., -0.5) is determined, drought events with different severity magnitudes are identified and constitute a sample for the selected time period. This sample contains different magnitudes in severity and different lengths in the duration, therefore, characterizes the distribution of different levels of drought (ranging from the mild, moderate to extreme conditions). In addition, the gamma distribution is applied to fit the distribution of different magnitudes of drought severity. To further confirm our results regarding drought risks under different levels of global warming, the threshold of -0.8 is also utilized, and the results derived from this threshold are similar to those from -0.5. Since the calculation of socioeconomic exposures to droughts is based on the variations of 50-year drought risk, similar changes in the drought risk will lead to analogical socioeconomic exposures. In other words, under a certain RCP scenario and for a certain warming level, drought risk changes determine the socio-economic exposures when employing the same dynamic population (and GDP) pathways. As a reference, we also analyze the socioeconomic exposures in the case when -0.8 is used as the threshold (Figs. R1-2). Compared with the results of the -0.5 threshold (Figs. 9-10), the overall characteristics of the drought exposures are mostly the same.

Furthermore, we also derive changes in drought risks for the 20-year or 100-year drought events to explore risk variations caused by different extents of drought (Figs. R3-4). Results shows that although the magnitudes of changes are different, they present quite similar spatial patterns.

These points have been added to the Discussion section of the revised manuscript (Page 20, Lines 26-29; Page 21, Lines 1-6), and Figs. R1-4 have been added in the supplementary (Fig. S5-8).

[Figure]

Figure R1 National population and GDP fraction exposing to more frequent severe droughts under the 1.5°C warming level (based on the -0.8 threshold)

[Figure]

Figure R2 National population and GDP fraction exposing to more frequent severe droughts under the 2.0°C warming level (based on the -0.8 threshold)

[Figure]

Figure R3 Projected changes of 20-, 50-, and 100-year joint return period of droughts under the 1.5°C warming level.

[Figure]

Figure R4 Projected changes of 20-, 50-, and 100-year joint return period of droughts between the 1.5°C and 2.0°C warming level.

*2. When calculating SPEI with Penman-Monteith-based PET, the term $(0.34u_2)$ in the equation is finally obtained through the ratio $r_s/r_a$ and represents the suggested reference crop surface (assuming a standard plant height of 0.12 m, affixed surface*

*resistance of 70 sm-1 and an albedo of 0.23). However, considering a distinct vegetation response to elevated $CO_2$ as simulated in the fully coupled climate models, it is important to point out that some of the assumptions that underlie the computation of PET (and thus SPEI) are incorrect (or at least the projected drought is not so severe) under conditions of changing $CO_2$ concentrations (Greve et al., 2019, ERL; Yang et al., 2018, NCC; Roderick et al., 2015, WRR).The authors should at least discuss the potential impacts of the elevated $CO_2$ on their drought risk assessment in Section 4.*

Reply: Thanks for this suggestion. Sorry we did not consider the impacts of increasing $CO_2$ concentrations on PET (and thus SPEI) in our study. This will be discussed as follows:

When calculating potential evapotranspiration based on the reference crop Penman-Monteith model, surface resistance ($r_s$) is fixed to 70 s/m. However, according to recent studies (e.g., Roderick et al., 2015; Yang et al., 2018), an elevated [$CO_2$] environment can drive stomatal closure, increasing stomatal resistance and further increasing $r_s$. Subsequently, this increasing $r_s$ causes the decline in the potential evapotranspiration, especially across vegetated lands where the photo-synthetic rate is high. From this perspective, the neglect of increasing $r_s$ may overestimate future drying condition and corresponding drought risk changes to some extent. However, on the other hand, the increase in total leaf area with [$CO_2$] and growing-season length can cause countervailing decreases in $r_s$ (Greve et al., 2019). Overall, accurate and robust quantification of $r_s$ scaling with [$CO_2$] still needs additionally explicit work and substantial observed data. Though the impact of $r_s$ on the drought assessments deserves further studies, it is beyond the scope of this study. Therefore, the traditional method is used in this study to calculate PET.

This point has been discussed in the revised manuscript (Page 19, Lines 18-29).

Greve, P., Roderick, M., Ukkola, A. M., & Wada, Y. The Aridity Index under global warming. Environmental Research Letters, 2019.

Roderick, M. L., Greve, P., & Farquhar, G. D. On the assessment of aridity with changes in atmospheric $CO_2$. Water Resources Research, 51(7), 5450-5463, 2015.

Yang, Y., Roderick, M. L., Zhang, S., McVicar, T. R., & Donohue, R. J. Hydrologic implications of vegetation response to elevated $CO_2$ in climate projections. Nature Climate Change, 9(1), 44, 2019.

*3. Given the relative coarseness of the CMIP5 models, I think interpolation of the results (especially bilinearly interpolated P and PET to a common resolution before calculating SPEI with them) to 1 degree spatial resolution is not appropriate. A 2 degree*

*common grid would be better, and would avoid effectively making up data at the much finer resolution. The authors should at least discuss the impact of interpolation on their results in the main-text.*

Reply: We agree with the reviewer it may be more appropriate to re-grid the GCM outputs to 2° common grid. However, the spatial resolution of population and GDP used in this study is 0.5°×0.5°, which have to be upscaled to the same resolution of GCM outputs. But the 2° grid may be larger than the largest city in the world, thus, it is inappropriate to reflect the regional population and GDP exposures. Besides, some national territory areas are small, a finer resolution (e.g., 1°×1°) may be more appropriate to obtain reliable population and GDP exposure results at the national scale. The same spatial resolution has been used in other studies (e.g., Schneider et al., 2016; Li et al., 2018; Yang et al., 2019).

Nevertheless, in order to validate the rationality of interpolation to 1° spatial resolution, we also re-gridded the data to 2° grid and further re-conducted our studies (Figs. R5-6). Overall, there are only slight differences between the results of 1° and 2° resolution, confirming the robustness of our results.

These clarifications have been presented in the revised manuscript (Page 24, Lines 26-29; Page 25, Lines 1-9). Corresponding figures have been added in the supplementary (Figs. S15-16)

Yang, Y., Roderick, M. L., Zhang, S., McVicar, T. R., & Donohue, R. J. Hydrologic implications of vegetation response to elevated $CO_2$ in climate projections. Nature Climate Change, 9(1), 44, 2019.
Li, W., Jiang, Z., Zhang, X., Li, L., & Sun, Y. Additional risk in extreme precipitation in China from 1.5 C to 2.0 C global warming levels. Science Bulletin, 63(4), 228-234, 2018.
Schneider, D. P., & Reusch, D. B. Antarctic and Southern Ocean surface temperatures in CMIP5 models in the context of the surface energy budget. Journal of Climate, 29(5), 1689-1716, 2016.

[Figure]

Figure R5 Projected changes in the mean and standard deviation of SPEI under the 1.5°C (a) and between the 1.5°C and 2.0°C (b) warming target at 2° spatial resolution

[Figure]

Figure R6 Projected changes in drought duration and severity under the 1.5°C (a) and between the 1.5°C and 2.0°C (b) warming target at 2° spatial resolution

*Some specific parts need further clarification.*

*1. During the investigation regarding the exposures of population and GDP to droughts under three RCP scenarios at two warming levels, for example, under the RCP8.5 scenario (SSP5), the specific time when future warming reaches 1.5ºC or 2ºC under RCP 8.5 can be different (from Fig 1), population and GDP can also possibly differ in two climates. From Line 17-Line 25 (Page 11), did the author suggest that the dynamic of population and GDP under RCP 8.5 at two warming climates was also considered using the multi-year average? If so, in section 3.4 about population and GDP exposure from increasing drought risks, it was concluded that a large percentage of population and GDP will be exposed to increasing drought risk. The drought risk has been increasing with warming climate, let's say if population and GDP have been increasing with time, then which one contributes to the increasing exposures, the increasing population or the increasing drought risks, I think this is a key question that authors should clarify when assessing the socioeconomic exposure.*

Reply: Thanks for this comment and sorry for the confusion of methodology of exposure analysis. We think the use of population and GDP corresponding to warming level periods instead of a single year (i.e. 2005 or 2100) which have been used by some earlier studies (e.g., Peters, 2016; Park et al., 2018; Liu et al. 2018a) may be more appropriate. The dynamic characteristics are considered as differences in population (and GDP) between the fixed 30-year 1.5°C and 2.0°C warming periods, and can be reflected by the multi-year average during warming climates to some extent (Table R1). In this way, variations in population (and GDP) and variations in drought risks can both lead to drought exposures changes. To further analyze their respective contributions, we rephrase the details as follows:

At the 1.5°C warming climate, there are around 88% of global landmasses being exposed to increasing drought risks, which correspond to 1386.9 million population (and 33311.1 billion USD) according to the average of the three RCPs from a global perspective. At the 2.0°C warming level, though there are still 88% of the global land areas being exposed to increasing drought risks, the affected population (and GDP) will soar to 1538.2 million (and 72852.2 billion USD). In this light, the increase in population (and GDP) contributes to the increasing exposures. Therefore, it is more appropriate to incorporate the dynamic population (and GDP) into exposure calculating processes.

When further investigating the affected population (and GDP) between the two warming climates, the role of drought risk changes should also pay attention. Specifically, though the percentage of landmasses with increasing drought risks stay unchanged for both the 1.5°C and 2.0°C warming climates (both approximately 88%), the magnitudes of risk changes are different. For instance, drought risks will double across around 58% of the global landmasses at the 1.5℃ warming level, while the same drought risks will occur over 67% of the global landmasses at the 2.0℃ warming level. Those differences in the magnitudes of drought risk changes can definitely bring about divergent impacts to local population and economy.

Related information have been clarified in the revised manuscript (Page 21, Lines 26-29; Page 22, Lines 1-24).

Liu, W., Sun, F., Lim, W. H., Zhang, J., Wang, H., Shiogama, H., and Zhang, Y.: Global drought and severe drought-affected populations in 1.5 and 2 °C warmer worlds. Earth Syst. Dynam., 9:267-283, 2018a.

Park, C. E., Jeong, S. J., Joshi, M., Osborn, T. J., Ho, C. H., Piao, S., and Kim, B. M.: Keeping global warming within 1.5° C constrains emergence of aridification. Nat. Clim. Change, 8(1), 70, 2018.

Peters, G. P.: The best available science to inform 1.5 C policy choices. Nat. Clim. Change, 6(7), 646. https://doi.org/10.1038/nclimate3000, 2016.

Table R1 Global population and GDP at the 1.5℃ and 2.0℃ warming climates

|  | SSP126 | SSP425 | SSP585 |
|---|---|---|---|
| **1.5°C-population (million)** | 1516.9 | 1553.5 | 1510.8 |
| **2.0°C-population (million)** | 1666.7 | 1731.2 | 1603.1 |
| **1.5°C-GDP (billion USD)** | 35875.0 | 34244.0 | 35668.5 |
| **2.0°C-GDP (billion USD)** | 116991.1 | 56271.6 | 58916.2 |

*2. Page 11, Line 13-16, how is the ratio of the recalculated recurrence frequency calculated and why a less than 1.0 ratio suggests worrisome drought condition. Need further clarification.*

Reply: Sorry for the confusion. The ratio of the re-calculated recurrence frequency is based on the joint probability distribution functions. Taking the 50-year drought events as an example, we first determine the magnitudes (duration and severity) of the 50-year drought events in the historical period. Then we input the determined magnitudes of the 50-year drought events into the future joint distribution functions, recalculate the joint recurrence frequencies and convert them into new return period at the 1.5℃ and 2.0℃ warming climates. The ratio is then calculated by dividing the new return period in the 2.0℃ warming future by the new return period in the 1.5℃ warming. A ratio less than 1.0 suggests that the new return period in 2.0℃ warming climates further reduces compared to that in 1.5℃ warming level, which means that reference drought events are more common under the 0.5℃ warming impacts.

In detail, if the recurrence frequency of the 50-year event increases at the 1.5℃ warming climate, the joint return period will decrease (e.g., become 30-year event); if the recurrence frequency of the 50-year event increases at even larger magnitudes at the 2.0°C warming climate, the joint return period will further decrease (e.g., become 20-year event). The ratio is then calculated by dividing the re-calculated joint return period in the 2.0°C warming level by that in the 1.5°C warming level (i.e., 20/30). Since drought events will become more frequent with additional 0.5°C warming, it implies worrisome conditions.

The information above has been clarified in Section 2.5 of the revised manuscript (Page 11, Lines 19-27).

*3. Page 12 section 3.1 projected changes in dryness, the author used SPEI and the run theory to define drought event, and the title of the manuscript is about the global drought, why would authors use SPEI to explain the dryness instead of using the defined event to study the changes in global drought for consistency.*

Reply: Thanks for this comment.
It should be noted that drought variations are different from the dryness condition under climate warming. Specifically, drought events are defined as abnormally dry conditions but cannot be used directly to explain the dryness. In other words, the projected dryness can lead to deteriorated drought conditions characterized by more frequent, longer, and more severe events, but not the other way around. Therefore, before performing drought evaluation under the rising temperature, there is a need to assess the projected climate dryness by using the drought index (i.e., SPEI). Consequently, we designed the projected changes in dryness in section 3.1 using SPEI and analyzed subsequent drought events changes in section 3.2. This framework is also consistent with previous studies (Ayantobo et al., 2017; Lehner et al., 2017). Following this procedure, the projected climatic water budget as well as the subsequent drought changes can be considered as a consequence of global warming.

Ayantobo, O.O., Li, Y., Song, S., Yao, N.: Spatial comparability of drought characteristics and related return periods in mainland China over 1961-2013. J. Hydrol., 550, 549-567, 2017.
Lehner, F., Coats, S., Stocker, T. F., Pendergrass, A. G., Sanderson, B. M., Raible, C. C., and Smerdon, J. E.: Projected drought risk in 1.5°C and 2°C warmer climates. Geophys. Res. Lett., 44: 7419-7428, 2017.

*4. Page 15 Line 28-29, whether the fraction of drought-affected population (or GDP) divided by total population (or GDP) can be a fairer and more impartial assessment is really hard to say given the fact that this method seems to cover up some most drought-affected countries, like the United States and China.*

Reply: Sorry for the confusion of the presentation. Instead of using the absolute value of population (and GDP) to assess the nation-wide drought exposures, we apply the nation-wide population (and GDP) fraction. That is, for a country (e.g., the United States), the fraction of drought-affected population (and GDP) divided by the total population (and GDP) of this country is employed as the indicator. Therefore, the most drought-affected countries are presented by high fractions. Moreover, the utilization of the fraction rather than the absolute value of nation-wide population (and GDP) can avoid covering up badly drought-affected countries where the national population (or GDP) are small (or low) regarding the world level.

This point has been rephrased in the revised manuscript in Page 16, Lines 1-6.

*5. Generally, in the discussion of either the magnitude of drought or the socioeconomic exposures of droughts, the differences between two warming targets are highlighted, however, the differences among three RCP scenarios are barely discussed in the manuscript. It makes me doubt the reason and necessity of using three RCP scenarios since they present almost similar variations under two warming targets. This issue might be even obvious in Fig 9 and 10, for example, in Fig 9, under RCP 4.5, population and GDP suggest 100% exposure to drought in Australia, which drops to 90% under RCP 8.5. Possible reasons and texts are needed here.*

Reply: Thanks for this suggestion. We give a rough discussion regarding the RCP uncertainty in Section 4 (Page 21, Lines 1-10). Though the three RCP scenarios present to some extent similar variations in terms of projected dryness patterns, there are still discernable differences in the projected drought risks and drought-affected exposures, especially when the warming increasing from the 1.5°C to the 2.0°C warming level (Fig. 8). Moreover, these differences will become more evident at the national scale (e.g., Figs S3-4). This will be explained as follows:

It is well-known that the warming trajectories are dependent on RCP scenarios. In other words, different RCP scenarios correspond to various temperature levels for the fixed time period. However, this study fixed the warming level. It can be expected that the differences among RCP scenarios are largely reduced. Nevertheless, the complex circulation system can still result in some differences in hydro-meteorological variables (e.g., precipitation, wind speed and relative humidity) among RCP scenarios, even at the same warming level, because they are not linearly related to the warming temperature. Since drought conditions are evaluated by using multiple hydro-meteorological variables, those differences at the same warming level can lead to variations in drought evolutions. Comparing to the middle and low emission pathway scenarios (RCP2.6 and RCP4.5), the high emission pathway scenario (RCP8.5) usually reaches the warming level at earlier time periods during which the greenhouse gas concentrations are relatively low. In this light, the projected drought conditions and drought-affected population (and GDP) can even be slightly less severe under RCP8.5, in contrast to situations under RCP 4.5 or RCP2.6. Therefore, it is not a surprise that under RCP 4.5, population (and GDP) suggest 100% exposure to drought in Australia, while it is smaller (99.8%) under RCP 8.5.

These points have been discussed in the revised manuscript (Page 23, Lines 14-29)

*6. Not sure whether section 3.5 is necessary since similar conclusions have been achieved in Fig 7 and 8, and these typical countries can just be used for further explanations in section 3.3. Besides, additional explanations for Fig 7g and Fig 8g are very necessary.*

Reply: Thanks for the comment. Section 3.3 presented the global drought risk changes at grid scales; while we find for assessment at the national scale, spatially aggregating mean changes are more helpful than per-grid cell changes to indicate the risk of a particular land fraction being impacted by climate change (Fischer et al., 2013). Therefore, we investigated more thoroughly the drought-affected land fractions (Figs. 11-12) by using a binning method (Page 16, Lines 26-29) to present spatially-aggregated mean changes for eight drought-prone countries in Section 3.5. Besides, section 3.5 calculated population (and GDP) exposing to increasing drought risks at different levels (e.g., <5, 5-10, 10-20, etc.) (Figs. S3-4), which can provide more systematic exposure information than those in section 3.4 which only counts population (and GDP) exposing to increasing drought risks as a whole.

In addition, with regards to Figs 7g-8g, they actually present the world land fraction subject to drought risk changes of different magnitudes under three RCPs. Specifically, for an individual climate model output, we calculate the land fraction using the ratio of grid counts located at certain extent (e.g., <5) divided by the world land grid counts (excluding Antarctic). Each box in Figs 7g-8g is stemmed from the 13 climate models results and the circle in each box represents the multi-model ensemble median results. According to Fig. 7g, around 88% of global landmasses (presented by smaller than 50-year return period) will be subject to more frequent reference droughts. In terms of Fig. 8g, more frequent droughts (indicated by less than 1 ratio) will occur over 71% of continental areas in 2.0°C warming level compared to 1.5°C warming. This point will be added in Section 3.3 of the revised manuscript.

This issue has been rephrased in the revised manuscript (Page 17, Lines 20-11) and the figure captions has been added in Figs. 7g-8g.

*Minor suggestions.*

*1. Citation of Fig. 3 somewhere between lines 21 and 22 in Page 12.*

Reply: Thanks. This has been added in the revised manuscript (Page 12, Line2).

*2. Writing in the manuscript should be more concise in the data and method section, e.g. Page 6 line 7, use surface maximum, mean, minimum air temperature to avoid repeat.*

Reply: Thanks and this has been revised in the manuscript.

*3. Table 2, extreme drought instead of extremely drought*

Reply: Thanks. This has been revised.

Replies to Referee #2

**Projected increases in magnitude and socioeconomic exposure of global droughts in 1.5 °C and 2 °C warmer climates**

Lei Gu, Jie Chen, Jiabo Yin, Sylvia C. Sullivan, Hui-Min Wang, Shenglian Guo, Liping Zhang, and Jong-Suk Kim

We thank the anonymous reviewer for the constructive comments and suggestions. We have provided detailed responses to each comment below and will revise the manuscript accordingly. For clarity, comments are given in *italics*, and our responses are given in plain text.

Authors' responses

**Legend**

*Reviewer's comments*

Authors' responses

*General Comments*

*In summary, motivated by the 2015 Paris Agreement proposal, this manuscript calculated global 3-month Standardized Precipitation Evapotranspiration Index (SPEI-3) based on 13 CMIP5 GCM simulations under three RCP scenarios (RCP2.6/4.5/8.5) during 1976-2100, quantified changes in global drought duration, severity and occurrence under a bivariate framework, and analyzed the drought exposures of populations and regional GDP under 3 shared socioeconomic pathways (SSPs) in future 1.5 and 2-degree warming worlds. Generally, this well written manuscript is able to portray drought evolution with different warming trajectories and provides information for climate adaptation strategies. Here I list several questions below and suggest acceptance of the manuscript after minor revision.*

We appreciate that the reviewer is favor of our manuscript. Please find our specific responses below.

*Minor Comments*

*(1) P7, L19-20. How do you determine the year in which a specific warming target is achieved? I suppose you select the median year of the 30-year period with surface*

*temperature closest to the warming target for each RCP (not for each RCP-GCM combination), so that the reaching year is the same for all 13 GCMs under a prescribed RCP scenario. The authors should clarify this.*

Reply: Sorry for that we did not clearly clarify this point. Yes, the period is determined based on multi-model ensemble mean of temperature. Thus, the reaching year is the same for all 13 GCMs under a specific RCP scenario. However, instead of using median year of the 30-year period, we used the 30-year running-mean. In other words, we selected the 30-year period with mean temperature closest to the warming target for each RCP.

This has been clarified in the revised manuscript (in Section 2.2, Page 7, Lines 19-22).

*(2) Figure 1a. I've noticed that the determined years under both scenarios (RCP2.6 and RCP8.5) are the first year (2020) of the whole period. Is it possible that for some GCM future projections, 1.5-degree warming (or even higher) has already been reached even at the beginning of the simulation period? If so, maybe it could affect the results in this manuscript.*

Reply: Thanks for this comment. We acknowledge that a few individual projections among the multi-model ensemble slight exceed the 1.5-degree warming at the beginning of the simulation period (i.e. BNU-ESM, CanESM2, GFDL-CM3 and MIROC-ESM-CHEM; Table R1). We will discuss this comment as follows:

To fully consider the robustness of the results, we use the warming level of multi-model ensemble mean to serve as the warming trajectory. Firstly, comparing to the method of determining warming level by individual model output, the use of multi-model ensemble mean method involves more future projections/GCMs and thus guarantees the reliability of the conclusions (Chen et al., 2011; Mehran et al., 2014). This multi-model ensemble mean method is also consistent with some previous studies (Liu et al., 2018a, 2019; Su et al., 2018). Secondly, the application of the multi-model ensemble mean method keeps the consistency of the sample size under each RCP and for each warming level. This can exclude the differences originated from the sample size when assessing different warming level impacts or evaluating RCP uncertainty. It is true that different warming level calculating methods can result in divergent model ensembles and may thus affect the results. For example, some studies (Sanderson et al., 2017; Lehner et al., 2017) used single model to conduct climate warming impact assessments, while some studies (James et al., 2017; Thober et al., 2018) employed pooled future projections (i.e. 1.5/2.0°C) to perform analyses without considering RCP discrepancies. Future studies may explore the impacts of different warming level calculation methods, but it is beyond the scope of the current study.

This has been added in the Discussion Section Page 24, Lines 8-25.

Table R1 Models with global warming higher than 1.5°C in the 2006 year (°C)

| MODEL | RCP2.6 | RCP4.5 | RCP8.5 |
|---|---|---|---|
| BNU-ESM | 1.503 | 1.583 | 1.540 |
| CanESM2 | 1.594 | 1.479 | 1.692 |
| GFDL-CM3 | 1.720 | 1.734 | 1.741 |
| MIROC-ESM-CHEM | 1.646 | 1.500 | 1.643 |

Chen, J., Brissette, F. P., Poulin, A., and Leconte, R.: Overall un- certainty study of the hydrological impacts of climate change for a Canadian watershed, Water Resour. Res., 47, W12509, https://doi.org/10.1029/2011wr010602, 2011.

James, R., Washington, R., Schleussner, C. F., Rogelj, J., & Conway, D. (2017). Characterizing half‐a‐degree difference: a review of methods for identifying regional climate responses to global warming targets. Wiley Interdisciplinary Reviews: Climate Change, 8(2), e457.

Lehner, F., Coats, S., Stocker, T. F., Pendergrass, A. G., Sanderson, B. M., Raible, C. C., and Smerdon, J. E.: Projected drought risk in 1.5°C and 2°C warmer climates. Geophys. Res. Lett., 44: 7419-7428, 2017.

Liu, W., Sun, F., Lim, W. H., Zhang, J., Wang, H., Shiogama, H., and Zhang, Y.: Global drought and severe drought-affected populations in 1.5 and 2 °C warmer worlds. Earth Syst. Dynam., 9:267-283, 2018a.

Liu, W., & Sun, F. Increased adversely-affected population from water shortage below normal conditions in China with anthropogenic warming. Science Bulletin, 64(9), 567-569, 2019.

Mehran, A., AghaKouchak, A., and Phillips, T. J.: Evaluation of CMIP5 continental precipitation simulations relative to satellite- based gauge-adjusted observations, J. Geophys. Res.- Atmos., 119, 1695–1707, https://doi.org/10.1002/2013jd021152, 2014.

Sanderson, B. M., Xu, Y., Tebaldi, C., et al.: Community climate simulations to assess avoided impacts in 1.5 and 2 °C futures. Earth Syst. Dynam., 8, 827-847, https://doi.org/10.5194/esd-8-827-2017, 2017.

Su, B., Huang, J., Fischer, T., Wang, Y., Kundzewicz, Z. W., Zhai, J., and Tao, H.: Drought losses in China might double between the 1.5° C and 2.0° C warming. P. Natl. Acad. Sci. USA., 115(42), 10600-10605, 2018.

Thober, S., Kumar, R., Wanders, N., Marx, A., Pan, M., Rakovec, O., ... & Zink, M. (2018). Multi-model ensemble projections of European river floods and high flows at 1.5, 2, and 3 degrees global warming. Environmental Research Letters, 13(1), 014003.

*(3) Figure 9, 10, S2 and Discussion Section. There are several countries (e.g. the United States) will experience a decrease in POP and GDP fraction exposing to more frequent severe droughts under the 2-degree warming level compared to 1.5-degree. I will be appreciated if the author could provide possible reasons considering the increasing drought risks in these countries.*

Reply: Thanks for this suggestion. Actually, countries that experience a decrease in POP and GDP exposure fraction under the 2°C warming can be attributed to the decreasing land fraction exposing to more frequent droughts. Here, we listed some example countries in Table R2 to analyze the reasons. We will add more analysis and revised clarifications in the revised manuscript (Section 3.4) as follows:

It should be noted that when climate warming climbing from 1.5°C to 2.0°C, there are some spatial heterogeneity with regards to drought exposures variations. Specifically, drought exposures for some countries (i.e., Canada) can be slightly decreased in 2°C warming level compared to 1.5°C warming level. This decrease in POP and GDP exposure fraction can be attributed to the decreasing land fraction exposing to more frequent droughts. For example, the land fraction suffering more frequent severe droughts in Canada will decrease (-12.77%) in 2.0°C warming level comparing to 1.5°C warming under RCP2.6 scenario. In other words, the additional 0.5°C warming will not lead to drought risk deterioration globally, partly due to the increasing column precipitable water with warming environment (Dong et al., 2019; Yin et al., 2019), although it holds for the majority of global land masses. Anyway, the spatial heterogeneity should be paid attention especially when assessing the climate change impacts on extreme events at regional or local scales (Liu et al., 2018b).

This has been added in the revised manuscript (Page 17, 5-18). The table has been added in the supplementary (Table S2).

Table R2 Several countries suffering decreasing POP and GDP exposure

| | Land fraction exposing to more frequent droughts | | | | | | | | |
| | 1.5°C | | | 2°C | | | 2-1.5°C | | |
| Country | RCP2.6 | RCP4.5 | RCP8.5 | RCP2.6 | RCP4.5 | RCP8.5 | RCP2.6 | RCP4.5 | RCP8.5 |
|---|---|---|---|---|---|---|---|---|---|
| Canada | 68.35% | 68.82% | 71.56% | 55.58% | 63.36% | 66.92% | -12.77% | -5.46% | -4.63% |
| United States | 86.17% | 78.93% | 86.89% | 85.44% | 80.02% | 79.84% | -0.72% | 1.08% | -7.05% |
| Colombia | 85.71% | 84.62% | 79.12% | 75.82% | 93.41% | 85.71% | -9.89% | 8.79% | 6.59% |
| Japan | 62.16% | 56.76% | 62.16% | 59.46% | 62.16% | 62.16% | -2.70% | 5.41% | 0.00% |
| | Population (million) | | | | | | | | |
| | 1.5°C | | | 2°C | | | 2-1.5°C | | |
| Country | SSP1 | SSP2 | SSP5 | SSP1 | SSP2 | SSP5 | SSP1 | SSP2 | SSP5 |
| Canada | 7.97 | 7.91 | 8.27 | 10.50 | 8.95 | 9.57 | 2.53 | 1.04 | 1.30 |
| United States | 59.13 | 58.82 | 60.75 | 73.56 | 64.86 | 68.20 | 14.43 | 6.05 | 7.45 |
| Colombia | 9.41 | 9.67 | 9.33 | 10.20 | 10.84 | 9.88 | 0.79 | 1.17 | 0.56 |
| Japan | 17.82 | 17.63 | 18.12 | 15.48 | 16.53 | 17.95 | -2.34 | -1.11 | -0.17 |
| | GDP (billion USD, 2010price PPP) | | | | | | | | |
| | 1.5°C | | | 2°C | | | 2-1.5°C | | |
| Country | SSP1 | SSP2 | SSP5 | SSP1 | SSP2 | SSP5 | SSP1 | SSP2 | SSP5 |
| Canada | 373.60 | 373.74 | 398.99 | 719.26 | 499.22 | 563.41 | 345.67 | 125.48 | 164.41 |
| United States | 3639.14 | 3517.35 | 3759.94 | 6699.26 | 4554.66 | 5118.33 | 3060.12 | 1037.32 | 1358.38 |

| | | | | | | | | | |
|---|---|---|---|---|---|---|---|---|---|
| **Colombia** | 192.32 | 184.47 | 191.51 | 617.84 | 296.93 | 311.37 | 425.52 | 112.46 | 119.86 |
| **Japan** | 575.07 | 553.76 | 590.57 | 873.40 | 620.73 | 730.21 | 298.33 | 66.97 | 139.63 |

**Population Exposure Fraction**

| | 1.5°C | | | 2°C | | | 2-1.5°C | | |
|---|---|---|---|---|---|---|---|---|---|
| **Country** | **SSP126** | **SSP245** | **SSP585** | **SSP126** | **SSP245** | **SSP585** | **SSP126** | **SSP245** | **SSP585** |
| **Canada** | 99.25% | 98.88% | 98.77% | 99.14% | 98.32% | 98.70% | -0.11% | -0.56% | -0.07% |
| **United States** | 99.84% | 99.60% | 99.85% | 99.82% | 99.85% | 99.84% | -0.02% | 0.26% | -0.01% |
| **Colombia** | 84.83% | 77.64% | 46.20% | 57.22% | 98.90% | 80.06% | -27.61% | 21.26% | 33.87% |
| **Japan** | 99.72% | 97.95% | 99.78% | 72.89% | 98.65% | 99.78% | -26.82% | 0.70% | 0.00% |

**GDP Exposure Fraction**

| | 1.5°C | | | 2°C | | | 2-1.5°C | | |
|---|---|---|---|---|---|---|---|---|---|
| **Country** | **SSP126** | **SSP245** | **SSP585** | **SSP126** | **SSP245** | **SSP585** | **SSP126** | **SSP245** | **SSP585** |
| **Canada** | 99.26% | 98.89% | 98.79% | 99.15% | 98.34% | 98.72% | -0.11% | -0.55% | -0.07% |
| **United States** | 99.84% | 99.59% | 99.85% | 99.82% | 99.85% | 99.84% | -0.02% | 0.26% | -0.01% |
| **Colombia** | 84.85% | 77.67% | 46.27% | 57.27% | 98.90% | 80.09% | -27.58% | 21.23% | 33.82% |
| **Japan** | 99.72% | 97.95% | 99.78% | 72.89% | 98.65% | 99.78% | -26.82% | 0.70% | 0.00% |

Dong, W., Lin, Y., Wright, J. S., Xie, Y., Yin, X., & Guo, J. Precipitable water and CAPE dependence of rainfall intensities in China. Climate Dynamics, 52(5-6), 3357-3368, 2019.

Liu, W., Lim, W. H., Sun, F., Mitchell, D., Wang, H., Chen, D., ... & Fischer, E. Global freshwater availability below normal conditions and population impact under 1.5 and 2 C stabilization scenarios. Geophysical Research Letters, 45(18), 9803-9813, 2018b.

Yin, J., Gentine, P., Guo, S., Zhou, S., Sullivan, S. C., Zhang, Y., ... & Liu, P. Reply to 'Increases in temperature do not translate to increased flooding'. Nature communications, 10(1), 1-5, 2019.

*(4) P10, L26. Eq. (5) should be Eq. (11)?*

Reply: Sorry for the mistake. Eq. (5) will be corrected as Eq. (11) in the revised manuscript.

*(5) As the authors mentioned in Section 2.1 that RCP2.6 is associated with SSP1, I suggest the author use SSP126 instead of RCP2.6 when talking about future drought risks. Same with SSP245 and SSP585.*

Reply: Thanks for this suggestion. This has been addressed in revise the statement in the manuscript and corresponding figures (Figs. 7-12; 14-15) has also been revised.

*(6) Relative to huge gaps in drought characteristics for two warming targets, results under three RCP scenarios seems to have few differences (e.g. Figure 6). Maybe the authors could explain the reason in Discussion Section, or explore the possible causes*

*in future studies.*

Reply: Thanks for this insightful comment. We have added the discussion as follows in the revised manuscript (Page 23, Lines 14-29):

[revised manuscript text omitted]

